# PBFORMER: CAPTURING COMPLEX SCENE TEXT SHAPE WITH POLYNOMIAL BAND TRANSFORMER

## ABSTRACT

We present PBFormer, an efficient yet powerful scene text detector that unifies the transformer with a novel text shape representation **P**olynomial **B**and (PB). The representation has four polynomial curves to fit a text's top, bottom, left, and right sides, which can capture a text with a complex shape by varying polynomial coefficients. PB has appealing features compared with conventional representations: 1) It can model different curvatures with a fixed number of parameters, while polygon-points-based methods need to utilize a different number of points. 2) It can distinguish adjacent or overlapping texts as they have apparent different curve coefficients, while segmentation-based methods suffer from adhesive spatial positions. PBFormer combines the PB with the transformer, which can directly generate smooth text contours sampled from predicted curves without interpolation. A parameter-free cross-scale pixel attention (CPA) module is employed to highlight the feature map of a suitable scale while suppressing the other feature maps. The simple operation can help detect small-scale texts and is compatible with the one-stage DETR framework, where no postprocessing exists for NMS. Furthermore, PBFormer is trained with a shape-contained loss, which not only enforces the piecewise alignment between the ground truth and the predicted curves but also makes curves' position and shapes consistent with each other. Without bells and whistles about text pre-training, our method is superior to the previous state-of-the-art text detectors on the arbitrary-shaped CTW1500 and Total-Text datasets. Codes will be public.

## 1 INTRODUCTION

Scene text detection is an active research topic in computer vision and enables many downstream applications such as image/video understanding, visual search, and autonomous driving (Radford et al., 2021; Long et al., 2021; Reddy et al., 2020). However, the task is also challenging. One non-negligible reason is that the text instance can have a complex shape due to the non-uniformity of the text font, skewing from the photograph, and specific art design. Capturing complex text shapes needs to develop effective text representation. State-of-the-art methods roughly tackle this problem with two types of representations. One is the point-based representation, which predicts the points on the image space to control the shape of the points, including the Bezier control points (Liu et al., 2020) and polygon points (Zhang et al., 2021). The other produces segmentation maps. The map can describe the text of various shapes and can benefit from the prediction results at the pixel level (Liao et al., 2020; Zhu et al., 2021b).

Despite the good performance, both types of representation have limitations: 1) Points-based methods suffer from a fixed number of control points (Tang et al., 2022; Zhang et al., 2022b). Too few points cannot handle the highly-curved texts, while simply adding points will increase redundancy for most perspective texts. 2) Segmentation-based methods frequently fail in dividing adjacent texts due to ambiguous spatial positions. The produced segmentation map still needs post-processing and often requires extensive training data (Zhu et al., 2021b).

To address these limitations, we propose a novel representation, named **P**olynomial **B**and (PB). It has clear advantages compared with previous text representations. In particular, PB consists of four polynomial curves, each of which fits along a text's top, bottom, left, and right sides. First, the coefficients of PB are discriminative in the parameter space even if the two texts are very close in

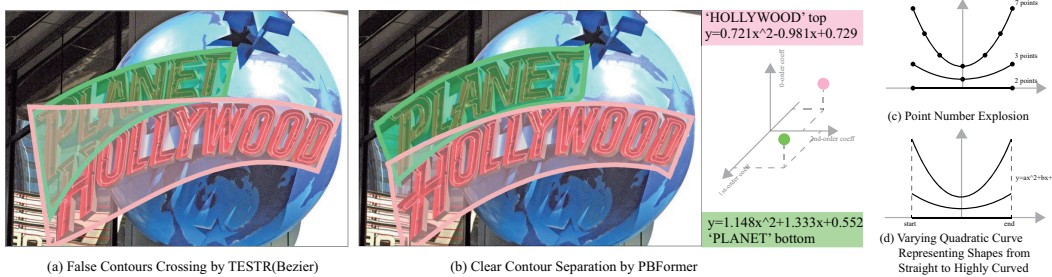

Figure 1: **Advantages of PB.** Comparing (a) and (b), PB divides adjacent texts more clearly than Bezier control points. (c) shows the number of output point increase gradually to represent shapes from straight to highly curved. (d) shows varying curve coefficients can handle dynamic shapes with two boundary variables.

the image space, as shown in Fig. 1. Second, PB is represented by the functions defined in the image space. We can directly compare the ground truth contour points with the sampled points from polynomial curves by re-sampling techniques. It differs from Bezier-curve-based methods that need to generate the intermediate representation, *i.e.*, "control points" for supervision. The loss defined by the control points cannot truly reflect how humans percept the shape. A small difference in control points will lead to a large shape difference.

Witnessing the great success in NLP (Vaswani et al., 2017), there has been a recent surge of interest in introducing transformers to vision tasks, including scene text detection. The current transformer-based text detectors are with two stages, such as FewBetter (Tang et al., 2022) and TESTR (Zhang et al., 2022b). We apply the proposed PB to the one-stage deformable DETR to improve the efficiency. In particular, we design parameter-free cross-scale pixel attention (CPA) module between the CNN feature and the transformer encoder-decoder layers. The CPA module first aligns the feature maps of different scales by enlarging all the feature maps to the same scale. Then it performs the cross-scale attention that highlights the feature value from a suitable scale while suppressing the other feature maps. With the scale-selective mechanism, our method becomes more compatible with the transformer decoders that do not have NMS for postprocessing. It implicitly suppresses the text proposals with incorrect scales, alleviating the learning burden of the transformer encoder-decoder layers. The features from CPA are effective to represent the shape of the text, two layers of transformer encoder-decoders are sufficient to detect the reasonable size of PB without NMS.

The transformer decodes each polynomial curve's $K$ coefficients and 2 boundary variables that determine the curve's definition domain. We uniformly sample the points on the predicted curves within the definition domain and compare them with the corresponding points on the ground truth polygon. Such a design supervises the curve piece by piece and can learn the curve shape and range consistently. In summary, the contribution of PBFormer is:

- A novel text representation called the Polynomial Band (PB) is proposed. PB can utilize a fixed number of parameters to capture the text instance with various curvatures. It also excels at distinguishing the spatially close text instances.

- A cross-scale pixel attention module is proposed. The module performs pixel-wise attention across the feature maps with different sizes. It implicitly highlights the text regions and enables the transformer to direct take all pixel-wise features as input.

- We design a shape-constrained loss function. The loss enforces the piece-wise supervision over the predicted curve and consistently optimizes the curve coefficients and definition domains.

Experiments on multi-oriented and curved text detection datasets CTW1500 (Liu et al., 2019) and Total-Text (Chng & Chan, 2017) demonstrate the effectiveness of our approach. Without any pre-training on large-scale text datasets, our method can achieve better results in terms of F-measure. Due to the lightweight network architecture, our method runs real-time and 4.4 × faster than other open-sourced transformer text detectors.

## 2 RELATED WORK

**Text representation.** Conventional text representation can be roughly divided into point-based and segmentation-based methods. With the need to capture more complex text shapes, the point-based methods gradually involve more points for text representation, including 2 points bounding box (Li et al., 2017), 4 points quadrilateral (Li et al., 2019; Sun et al., 2018), 16 points polygon (Zhang et al., 2020) or 30 points polygon (Zhang et al., 2020) changed by texts' length. ABC-Net (Liu et al., 2020) calculates Bezier control points (8 points), which is sufficient for most quadrilateral or slightly-curved texts but still suffers from highly-curved. One deficiency of point-based methods is that they usually predict a fixed number of points, which is hard to balance the performance between simple perspective texts and text instances with complex shapes (Zhang et al., 2022b). ESIR (Zhan & Lu, 2019) adopts a single polynomial that fits the text center line to rectify the image for recognition. However, they can only model horizontal texts and still need points to represent text contours. In contrast, the polynomial curves in polynomial band can handle various orientations and shapes, from a straight line to a round curve with fewer and a fixed number of parameters.

Segmentation-based representation can naturally handle complex-shaped texts due to pixel-wise description (Wang et al., 2019a; Liao et al., 2021; Wang et al., 2019b; Tian et al., 2019; Zhu et al., 2021b; Liao et al., 2020; 2022). However, they frequently fail to divide adjacent texts due to ambiguous spatial positions. Although CentripetalText (Sheng et al., 2021) tackles this problem by detecting a shrunk text mask and reconstructing the contour by shift map, it suffers from high computation complexity. The proposed polynomial band overcomes this problem by considering the global curve shape. Curve coefficients in parameter space can be easily separated even though two texts are close, making more clear bounds in the text crowding scenario.

**Text detection transformer.** There is a trend to equip the transformer (Vaswani et al., 2017) with scene text detection. Current methods (Tang et al., 2022; Zhang et al., 2022b) directly combine DETR variants (Carion et al., 2020; Zhu et al., 2021a) with point representation such as 16-point polygons or Bezier control points, and adopt the two-stage architecture for the ease of optimization. For example, FewBetter (Tang et al., 2022) first extracts segmentation maps by CNN-FPN (Lin et al., 2017) to show representative text regions, then samples feature points in each region and feed them into a transformer to further decode control points. TESTR (Zhang et al., 2022b) follows (Wang et al., 2020b;c) to detect the bounding boxes, then utilize a transformer to find the control points in each bounding box. They sacrifice efficiency due to the two-stage pipelines and destroy the simplicity of the detection transformer scheme. Our PBFormer inherits the single-stage simplicity and inserts a parameter-free cross-scale pixel attention module between the CNN feature and transformer encoder-decoder layers. With the attention module, per-pixel features are fed into the subsequent transformer without generating any proposals, making the whole architecture more efficient.

## 3 METHODOLOGY

The overall framework of PBFormer is illustrated in Fig. 2. Given an image with texts, PBFormer first employs a ResNet50 to produce the multi-scale feature maps, then feed the feature maps to a cross-scale pixel attention module to highlight the texts' context information. The enhanced feature maps are concatenated and fed to a lightweight transformer, predicting the PBs' parameters. PB utilizes four polynomial curves to represent the shape of the text instance. It is simple but effective to capture different forms of texts. In the training phase, we sample the dense points from the curves for PB parameters estimation according to both predicted curve coefficients and the domain variables. A shape-constrained loss is designed to supervise the curves piece by piece.

### 3.1 NETWORK ARCHITECTURE.

The network contains three modules: a ResNet-based CNN encoder, a cross-scale pixel attention module, and a lightweight deformable transformer. We now introduce more details about the attention module and the decoders in the transformer.

**Cross-scale pixel attention.** The motivation of cross-scale pixel attention is to highlight text features at the best scale and suppress the others. Such a selective mechanism is compatible with

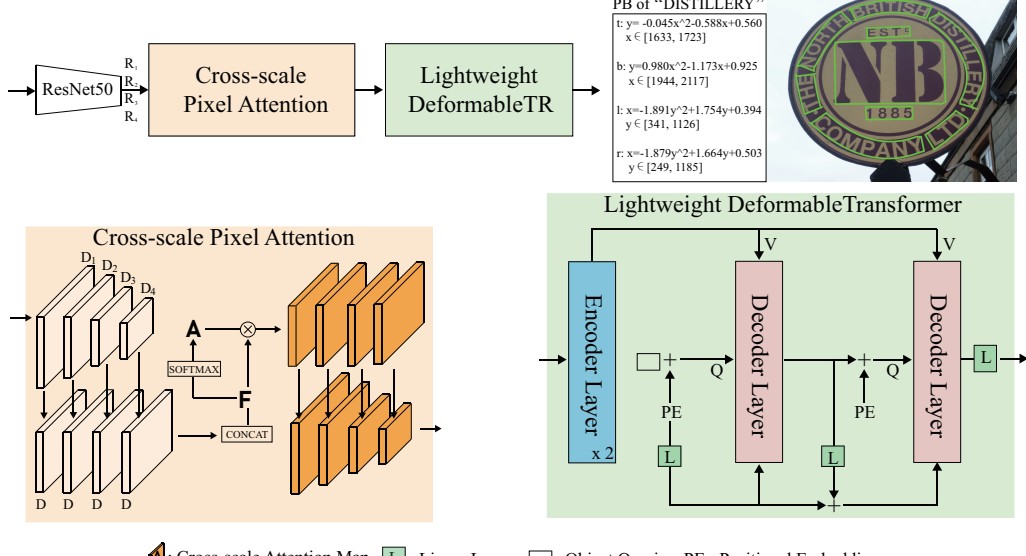

Figure 2: **Architecture of PBFormer.** The $\otimes$ and $+$ are element-wisely multiplication and addition, respectively. In the lightweight Deformable-Transformer, the object queries (white box) and Positional Embeddings (PE) are all learned parameters.

DETR-like detectors that do not have non-maximum suppression. It performs scale attention by comparing the values of the existing features among different scales with SoftMax. More details are illustrated in Fig. 2, the feature maps of a square image from the ResNet backbone are with the size $R_1, R_2, R_3, R_4$. We enlarge them to square feature maps with sizes $D_1, D_2, D_3, D_4$. Following Deformable-DETR, the four feature maps are transformed to have the same channels by $1 \times 1$ convolutions. Then we re-scaled them to have the same size $D$ and assemble them to obtain $\mathbf{F} \in \mathbb{R}^{D \times D \times C \times 4}$. After that, we use a SoftMax layer to compute the attention map for each pixel and each channel: $\mathbf{A}^{ijk} = \mathrm{softmax}([\mathbf{F}^{ijk1}, \mathbf{F}^{ijk2}, \mathbf{F}^{ijk3}, \mathbf{F}^{ijk4}]), \quad i \in [1, D], \quad j \in [1, D], \quad k \in [1, C]$. The attention map $\mathbf{A}$ and the feature map $\mathbf{F}$ have the same shape, *i.e.*, $\mathbf{A}, \mathbf{F} \in \mathbb{R}^{D \times D \times C \times 4}$. They multiply together to obtain the enhanced feature map $\mathbf{F}'$. We disassemble $\mathbf{F}'$ to four feature maps. All of them are with the size $D$ and the channel dimension $C$. We re-scale their size to be $D_1, D_2, D_3$ and $D_4$. It is noteworthy that the whole cross-pixel attention module is parameter-free, which brings no training burdens during gradient back-propagation.

**Lightweight deformable transformer.** The four feature maps from the cross-scale pixel attention module are flattened to be vectors. We concatenate them to a long vector, then feed the vector to the deformable transformer. To predict the parameters of PB, we reduce the layers of the standard transformer encoder and decoder from 6 to 2, which is sufficient to yield competitive results. In a deformable transformer, the reference points attend a small set of key sampling points nearby for each query, which are important to the deformable attention module. We adopt a coarse-to-fine strategy to generate the reference points for the two decoder layers. In the first decoder, we adopt rough 2-d reference points derived from the positional embedding via a linear projection. In the second decoder, we combine the same 2-d reference points with a 2-d vectors transformed from the output of the first decoder. In particular, the 2-d vectors encode the relative offsets according to the first decoder's learned non-local dependencies, which help to generate more reasonable reference points for the second decoder layer. After that, a 3-layer MLP generates the PB predictions over the entire image.

### 3.2  POLYNOMIAL BAND

We utilize four polynomial curves to represent the text instance's top, bottom, left, and right sides. The top and bottom boundaries are represented by $y = f^t(x)$ and $y = f^b(x)$:

$$
\begin{aligned}
y = f^t(x) = a_2^t x^2 + a_1^t x + a_0^t, \quad x \in \left[e_0^t, e_1^t\right], \\
y = f^b(x) = a_2^b x^2 + a_1^b x + a_0^b, \quad x \in \left[e_0^b, e_1^b\right],
\end{aligned}
\tag{1}
$$

where $(x, y)$ is the coordinate of a point on the boundary. $a_2^t, a_1^t, a_0^t, a_2^b, a_1^b, a_0^b$ are polynomial coefficients. $[e_0^t, e_1^t]$ and $[e_0^b, e_1^b]$ are range of $x$ variable.

One critical problem is that the polynomial curve is a *single-value function* which means one point in the definition domain has a unique value in the value domain. The functions of the curves along the horizontal direction cannot be used to represent the curves along the vertical direction. For example, as Fig. 2 shows, the text 'DISTILLERT's left (or right) side would not be represented by any $y = f(x)$. We utilize $x = f^l(y)$ and $x = f^r(y)$ to represent the left and right polynomial curves of the text instance:

$$
\begin{aligned}
x = f^l(y) = a_2^l y^2 + a_1^l y + a_0^l, \quad y \in \left[e_0^l, e_1^l\right], \\
x = f^r(y) = a_2^r y^2 + a_1^r y + a_0^r, \quad y \in \left[e_0^r, e_1^r\right],
\end{aligned}
\tag{2}
$$

where $[e_0^l, e_1^l]$ and $[e_0^r, e_1^r]$ define the range of $y$ variable.

**Output definition.** We use four polynomial curves $y = f^t(x), y = f^b(x), x = f^l(y), x = f^r(y)$ that denote a band to wrap a text instance. Thus, the output is a 20-tuple $\theta$ that consists of all polynomial coefficients and boundary variables in the form of:

$$
\theta = \left(a_2^t, a_1^t, a_0^t, e_0^t, e_1^t, a_2^b, a_1^b, a_0^b, e_0^b, e_1^b, a_2^l, a_1^l, a_0^l, e_0^l, e_1^l, a_2^r, a_1^r, a_0^r, e_0^r, e_1^r\right),
\tag{3}
$$

where $a_2^t, a_2^b, a_2^l, a_2^r \neq 0, e_0^t, ... \in [0, 1]$, and all of them are real numbers.

### 3.3 Loss Function

We leave the part about how to generate four ground truth point sets for the top, bottom, left and right sides from an original annotated polygon annotation in the Sec. 6.4. In this section, we introduce the shape-constrained loss to supervise the whole network. The network outputs $N$ different PB parameters for each image, while their correspondences to ground truth contours are unknown. In this section, we first introduce how to compute the similarity between the predicted PB and ground truth contour by shape-constrained loss, then provide the loss function for the whole image based on optimized correspondences solved by bipartite matching.

**The shape constraints for each curve.** We first revisit the curve fitting loss without constraints used in the lane detection (Liu et al., 2021; 2022). The ground truth fitting points of a top or bottom curve are given by:

$$
\hat{\mathcal{P}} = \{(\hat{x}_i, \hat{y}_i)\}_{i=0}^K, \quad \hat{x}_i = \hat{x}_0 + \frac{\hat{x}_K - \hat{x}_0}{K} i,
\tag{4}
$$

where the points are ordered from one end to the other, and the adjacent points for the top and bottom curves have the equal distance. The conventional fitting loss is:

$$
\mathcal{L}_{w/o}(\hat{\mathcal{P}}) = \sum_{i=0}^K \|\hat{y}_i - f(\hat{x}_i)\|_1 + \|e_0 - \hat{x}_0\|_1 + \|e_1 - \hat{x}_K\|_1.
\tag{5}
$$

The fitting loss for the left and right curves can be obtained by exchanging the $x$ and $y$ variables. However, such a fitting loss is unsuitable for detecting texts with diverse shapes and different positions. It has two limitations: (1) the predicted curve segment is not aligned with the ground truth fitting points piece-by-piece; (2) the shape and range of the curve are independently optimized. As demonstrated in Fig. 3, the conventional loss is not sensitive to the length of the curves, therefore the text detector tends to detect curves with inaccurate lengths.

We consider to impose the shape constraints. The points on the predicted curve are sampled according to both curve shape and range:

$$
\mathcal{P} = \{(x_i, f(x_i))\}_{i=0}^K, \quad x_i = e_0 + \frac{e_1 - e_0}{K} i.
\tag{6}
$$

Then we compare the predicted points $\mathcal{P}$ and ground truth points $\hat{\mathcal{P}}$:

$$
\mathcal{L}(\mathcal{P}, \hat{\mathcal{P}}) = \sum_{i=0}^K \|x_i - \hat{x}_i\|_1 + \|f(x_i) - \hat{y}_i\|_1.
\tag{7}
$$

Leveraging Eq. 7 in text detection encourages PB to reconstruct the correct length of the contours.

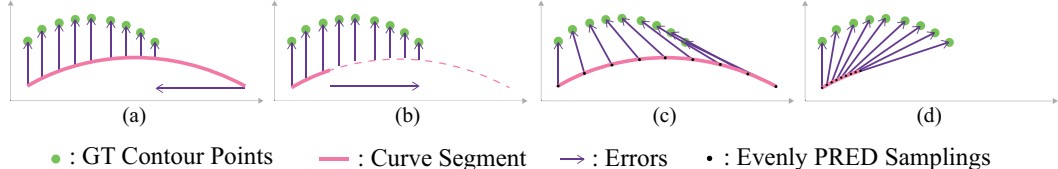

Figure 3: **Diagram of shape constraints.** All curve segments have the same curve coefficients. (a) and (c) have same ranges, so do (b) and (d). Without shape constraints, (a) and (b) show how to compare predicted curve segment with ground truth contour points. (c) and (d) illustrate the way with shape constraints.

**The bipartite matching for the whole image.** Let the network output of one image be $\mathcal{H} = \{h_j = (c_j, \theta_j)\}_{j=1}^N$, where $c_j$ is the confidence score indicating the possibility of a PB covering a text and $N$ is set to be larger than the maximum number of texts in an image. After sampling the points on the four curves according to Eq. 6, $\mathcal{H}$ can be further represented by: $\mathcal{H} = \{h_j = (c_j, \mathcal{P}_j^t, \mathcal{P}_j^b, \mathcal{P}_j^l, \mathcal{P}_j^r)\}_{j=1}^N$.

For bipartite matching, we pad the ground truth set $\hat{\mathcal{H}}$ with non-text instances to have a size $N$. The element having text instance is represented by $\hat{h}_j = (\hat{c}_j, \hat{\mathcal{P}}_j^t, \hat{\mathcal{P}}_j^b, \hat{\mathcal{P}}_j^l, \hat{\mathcal{P}}_j^r)$. In particular, $\hat{\mathcal{P}}_j^t, \hat{\mathcal{P}}_j^b, \hat{\mathcal{P}}_j^l, \hat{\mathcal{P}}_j^r$ are sampled points according to Eq. 4, while they are not need to be instantiated in the matching cost for non-text instances thus are set to be $\emptyset$. $\hat{c}_j$ is set to be 1 for the text and 0 for the non-text class. We formulate a bipartite matching problem to find an optimal injective function $g : \hat{\mathcal{H}} \to \mathcal{H}$, *i.e.*, $g(i)$ is the index of the PB assigned to fitting the $i$-th ground truth text:

$$g^* = \arg\min_g \sum_{j=1}^N \mathcal{L}^{fit}\left(\hat{h}_j, h_{g(j)}\right) + \mathcal{C}^{focal}\left(\hat{c}_j, c_{g(j)}\right), \tag{8}$$

where $\mathcal{L}^{fit}$ is the fitting loss and $\mathcal{C}^{focal}$ is the focal cost. The fitting loss compares the predicted contour and ground truth contour by using the loss defined in Eq. 7:

$$\mathcal{L}^{fit}\left(\hat{h}_j, h_{g(j)}\right) = \mathbf{1}_{\hat{c}_j>0}\left(\mathcal{L}(\hat{\mathcal{P}}_j^t, \mathcal{P}_{g(j)}^t) + \mathcal{L}(\hat{\mathcal{P}}_j^b, \mathcal{P}_{g(j)}^b) + \mathcal{L}(\hat{\mathcal{P}}_j^l, \mathcal{P}_{g(j)}^l) + \mathcal{L}(\hat{\mathcal{P}}_j^r, \mathcal{P}_{g(j)}^r)\right). \tag{9}$$

Then, the focal cost is defined as the difference between the positive and negative costs:

$$\mathcal{C}^{focal}\left(\hat{c}_j, c_{g(j)}\right) = \lambda \mathbf{1}_{\hat{c}_j>0}\left[-\alpha\left(1 - c_{g(j)}\right)^\gamma \log c_{g(j)} + (1 - \alpha) c_{g(j)}^\gamma \log\left(1 - c_{g(j)}\right)\right], \tag{10}$$

where $\alpha$ and $\gamma$ are the hyper-parameter for the focal loss. $\alpha$ is used to address the class imbalance, and $\gamma$ adjusts the rate at which easy examples are down-weighted. $\lambda$ adjusts the weight of the focal cost. The bipartite problem (Eq. 8) can be efficiently solved by the Hungarian algorithm.

**Overall Loss.** With the optimized $g^*$, the overall loss function is given by:

$$\mathcal{L}^{overall} = \sum_{j=1}^N \mathcal{L}^{fit}\left(\hat{h}_j, h_{g^*(j)}\right) + \mathcal{L}^{focal}\left(\hat{c}_j, c_{g^*(j)}\right), \tag{11}$$

where the $\mathcal{L}^{focal}\left(\hat{c}_j, c_{g^*(j)}\right)$ is the focal loss:

$$\mathcal{L}^{focal}\left(\hat{c}_j, c_{g^*(j)}\right) = \lambda\left[\mathbf{1}_{\hat{c}_j>0} - \alpha\left(1 - c_{g^*(j)}\right)^\gamma \log c_{g^*(j)} - \mathbf{1}_{\hat{c}_j=0}(1 - \alpha) c_{g^*(j)}^\gamma \log\left(1 - c_{g^*(j)}\right)\right]. \tag{12}$$

$\alpha$, $\lambda$ and $\gamma$ are the same with the ones in Eq. 10.

## 4 EXPERIMENTS

**Datasets.** CTW1500 (Liu et al., 2019) is a multi-oriented and curved scene text detection benchmark containing 1,000 training and 500 testing images. Annotations are based on the text-line level with fixed fourteen points. The majority of text instances are curved. Total-Text (Chng & Chan, 2017) is an another multi-oriented and curved scene text benchmark, while it consists of various text shapes such as multidirectional quadrilateral. It has 1255 training images and 300 testing images. Each instance is annotated by ten point text-line.

Table 1: Detection results on CTW1500 and Total-Text **without** pre-training on any text datasets. "Rep." denotes the method's output representation. "F.", "Prec.", "Rec." represent F-measure, Precision, and Recall. All the results are from their official codes and models. TESTR* means training without recognition branch.

| Method | Rep. | CTW1500 | | | | Total-Text | | | |
|---|---|---|---|---|---|---|---|---|---|
| | | F. | Prec. | Rec. | FPS | F. | Prec. | Rec. | FPS |
| PSENet | Seg | 78.0 | 80.6 | 75.6 | 3.9 | 78.3 | 81.8 | 75.1 | 3.9 |
| PAN | Seg | 81.0 | 84.6 | 77.7 | 39.8 | 83.5 | 88.0 | 79.4 | 39.6 |
| FCENet | Seg | 85.1 | 88.1 | 82.3 | 2.7 | 85.8 | 89.3 | 82.5 | 2.9 |
| ContourNet | Pts | 83.9 | 83.7 | 84.1 | 3.8 | 85.4 | 86.9 | **83.9** | 3.8 |
| TextBPN | Pts | 84.0 | 87.7 | 80.6 | 12.1 | 86.9 | 90.8 | 83.3 | 10.6 |
| TESTR* | Pts | 85.1 | 88.4 | 82.1 | 7.3 | 85.3 | 89.7 | 81.2 | 6.9 |
| TESTR | Pts | 85.3 | 87.9 | 82.8 | 5.6 | 85.6 | 90.7 | 81.1 | 5.3 |
| TESTR* | Bez | 84.7 | 87.9 | 81.8 | 7.3 | 86.3 | 90.3 | 82.6 | 6.9 |
| TESTR | Bez | 85.1 | 88.0 | 82.4 | 5.6 | 86.6 | 91.3 | 82.4 | 5.3 |
| PBFormer | PB | **87.0** | **89.6** | **84.5** | 24.7 | **87.1** | **92.1** | 82.6 | 24.6 |

Table 2: Detection results on CTW1500 and Total-Text **with** pre-training on text datasets. MLT, ST, ArT, and CST are abbreviations for MLT2017, SynthText, ArT 2019 and CurvedSynthText datasets. C+M+T means using a combination of CST, MLT, and Total-Text for pre-training.

| Method | Rep. | Ext. | CTW1500 | | | | Total-Text | | | |
|---|---|---|---|---|---|---|---|---|---|---|
| | | | F. | Prec. | Rec. | FPS | F. | Prec. | Rec. | FPS |
| 1 PSENet | Seg | MLT | 82.2 | 84.8 | 79.7 | 3.9 | 80.9 | 84.0 | 78.0 | 3.9 |
| PAN | Seg | ST | 83.7 | 86.4 | 81.2 | 39.8 | 85.0 | 89.3 | 81.0 | 39.6 |
| DB | Seg | ST | 83.4 | 86.9 | 80.2 | 22 | 84.7 | 87.1 | 82.5 | 32 |
| DB++ | Seg | ST | 85.3 | 87.9 | 82.8 | 26 | 86.0 | 88.9 | 83.2 | 28 |
| TextRay | Pts | ArT | 81.6 | 82.8 | 80.4 | 3.2 | 80.6 | 83.5 | 77.9 | 3.5 |
| DRRG | Pts | MLT | 84.5 | 85.9 | 83.0 | - | 85.7 | 86.5 | 84.9 | - |
| TextBPN | Pts | MLT | 85.0 | 86.5 | 83.6 | 12.2 | 87.9 | 90.7 | 85.2 | 10.7 |
| TESTR* | Pts | C+M+T | 86.6 | 90.8 | 82.8 | 7.3 | 86.2 | 92.4 | 80.7 | 6.9 |
| TESTR | Pts | C+M+T | 87.1 | **92.0** | 82.6 | 5.6 | 86.9 | **93.4** | 81.4 | 5.3 |
| ABCNet | Bez | CST | 81.4 | 84.4 | 78.5 | 6.8 | 84.5 | 87.9 | 81.3 | 6.9 |
| FewBetter | Bez | CST | 85.2 | 88.1 | 82.4 | - | **88.1** | 90.7 | **85.7** | - |
| TESTR* | Bez | C+M+T | 85.9 | 90.6 | 81.6 | 7.3 | 87.4 | 92.4 | 82.8 | 6.9 |
| TESTR | Bez | C+M+T | 86.3 | 89.7 | 83.1 | 5.6 | 88.0 | 92.8 | 83.7 | 5.5 |
| PBFormer | PB | CST | **88.0** | 90.6 | **85.4** | 24.7 | **88.1** | 93.2 | 83.5 | 24.6 |

**Evaluation Metrics.** We follow the standard metrics F-measures, recall, and precision to evaluate the performance. A prediction is considered as a true positive only when its IoU from the corresponding ground truth contour is larger than 0.5.

**Implementation Details.** The input image size is set to be $800 \times 800$ for training and testing. Loss coefficient $\alpha$, $\gamma$ and $\lambda$ are set as 0.25, 2 and 2. The fixed number of output $N$ is 300. In the cross-scale pixel attention module, $R_1, R_2, R_3, R_4$ are 100, 50, 25, 13, $D_1, D_2, D_3, D_4$ are set as 128, 64, 32, 16, and we set $D = D_2$. For training from scratch, the learning rate is set to be $1 \times 10^{-4}$ and decayed ten times at 7200 epochs, and the total number of training iterations is set as 9000 epochs. The training process takes about 2 days on 4 Tesla V100 GPUs with the image batch size of 14. For training with pre-training, we pre-train the model for 50 epochs, then fine-tune the model on CTW1500 and Total-Text by the same setting as training without pre-training states.

### 4.1 COMPARISON WITH THE STATE-OF-THE-ART METHODS

To demonstrate the effectiveness of our method, we take (1) point-based methods ABCNet (Liu et al., 2020), TextRay (Wang et al., 2020a), DRRG (Zhang et al., 2020), ContourNet (Wang et al., 2020c) and TextBPN (Zhang et al., 2021); (2) segmentation-based methods PSENet (Wang et al., 2019a), PAN (Wang et al., 2019b), DB (Liao et al., 2020), FCENet (Zhu et al., 2021b), and DB++ (Liao et al., 2022); and (3) recent transformer-based methods FewBetter (Tang et al., 2022)

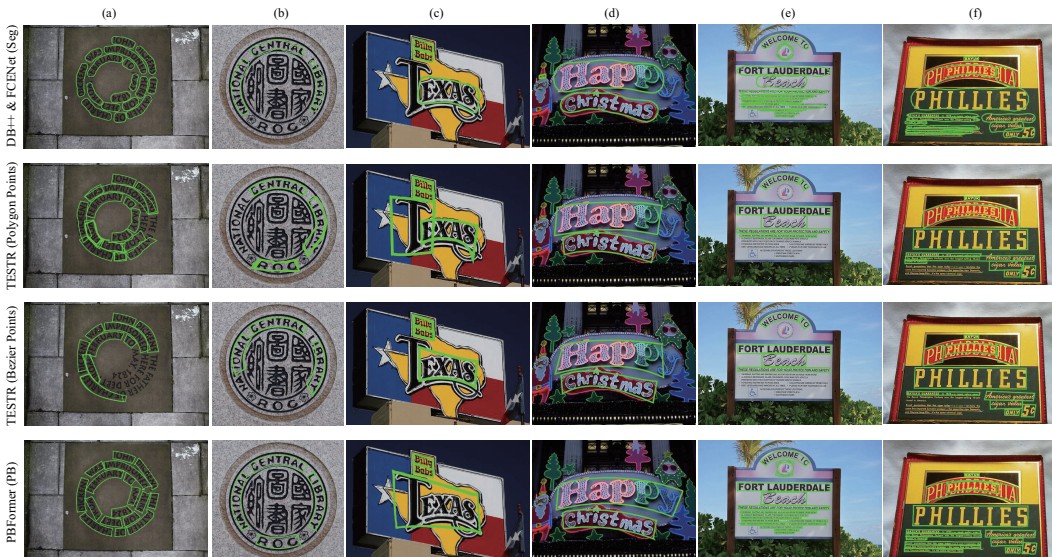

Figure 4: **Qualitative comparisons with previous SOTA on Total-Text and CTW1500.** Compared to DB++ and FCENet, our PBFormer predicts more compact and precise contours for crowded texts (the first two are DB++'s Total-Text detections, and the last four are FCENet's CTW1500 detections, because they did not release the model of another dataset). Compared to TESTR, PBFormer reduces false negatives and performs better for long and curved texts.

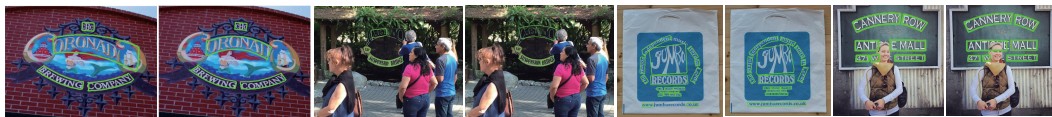

Figure 5: **Effect visualization of shape-constrained loss**. For each image pair, the left image shows the results with the fitting loss Eq. 7, and the right image is with the shape-constrained loss Eq. 11. With the shape-constrained loss, PBFormer outputs more complete contours.

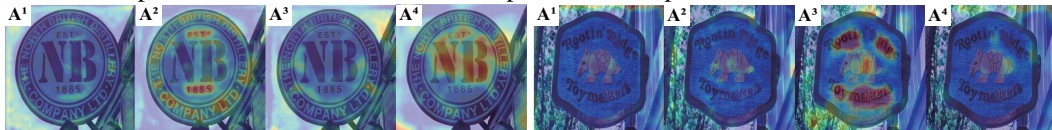

Figure 6: **Visualization of CPA's attention maps.** The left four images show CPA produces attention for the small texts at a swallow layer and the large texts at a deep layer. The right four image show attention is concentrated on one layer due to texts having similar sizes.

and TESTR (Zhang et al., 2022b). TESTR* means we use their official training codes, settings, and models but only set the recognition loss to zeros. For PBFormer, we will not use text character annotation for text detection training.

**Models trained from scratch.** As shown in Tab. 1, PBFormer establishes a new state-of-the-art of **87.0**% on CTW1500, which is **1.7**% better than previous best TESTR while achieving **4.4** × FPS. Moreover, PBFormer yields the best F-measure **87.1**% on Total-Text, which is **0.2**% better than previous best TextBPN while being **2.3** × FPS. Compared to TESTR, PBFormer also improves it by **0.8**% while keeping a **4.5** × FPS.

**Models with pre-training.** Since previous methods have different choices of pre-training datasets, we choose to use the CurvedSynthText as FewBetter did. As shown in Tab. 2, PBFormer can achieve better results from the model pre-training. On the CTW1500 dataset, our method achieves the best results. It outperforms previous best TESTR by **0.9**% in terms of F-measure and is **4.4** × faster than TESTR. On the Total-Text dataset, PBFormer also has the best performance. Compared to TESTR, PBFormer performs a **4.5** × FPS while being a slight **0.1**% higher F-measure. Without training TESTR's recognition branch, PBFormer's improvements are more obvious by **+1.4**% and **+0.7**%.

In addition, PBFormer yields **2.7**% and **2.1**% F-measure better than the previous best segmentation-based DB++ on CTW1500 and Total-Text, while keeping a very close FPS performance.

**Qualitative comparisons.** Considering crowded texts in Fig. 4(a),(e), and (f), PBFormer performs fewer false-negatives than TESTR and more accurate contours than DB++ and FCENet. When texts have very long shapes or have characters' large scale-changes, PBFormer detected more completed contours than DB++, FCENet, and TESTR, as Fig. 4(d) and (c) have shown.

Table 3: Comparison on Total-Text of the same network with different text representations.

| CPA | Ref. | Rep. | F. | Prec. | Rec. |
|---|---|---|---|---|---|
| - | - | PB | **85.8** | **90.2** | **81.9** |
| - | - | Pts | 82.8 | 88.4 | 77.8 |
| - | - | Bez | 83.9 | 89.8 | 78.7 |
| - | ✓ | PB | **86.0** | **90.5** | **81.9** |
| - | ✓ | Pts | 83.2 | 87.6 | 79.1 |
| - | ✓ | Bez | 84.2 | 89.7 | 79.4 |
| ✓ | ✓ | PB | **87.1** | **92.1** | **82.6** |
| ✓ | ✓ | Pts | 84.6 | 90.5 | 79.5 |
| ✓ | ✓ | Bez | 85.3 | 90.9 | 80.4 |

Table 4: CPA's influences and comparisons with FPN and ASF. Ext. means trainable parameters.

| Module | Ext. | F. | Prec. | Rec. |
|---|---|---|---|---|
| - | - | 86.0 | 90.5 | 81.9 |
| Enlarge | - | 85.2 | 88.9 | 81.8 |
| Attn. | - | 86.1 | 90.6 | 82.1 |
| CPA | - | **87.1** | **92.1** | **82.6** |
| ASF | ✓ | 85.3 | 90.9 | 80.4 |
| FPN | ✓ | 85.0 | 89.5 | 81.0 |

## 4.2 ABLATION STUDY

**Investigation of proposed components.** In this section, we investigate the effects of different text representations under the same network and validate the effects of CPA and refined reference points.Tab. 3 shows the following conclusions: (1) PB outperforms polygon points and Bezier control points in every network configuration. In Fig. 12, we also show that PB can distinguish individual texts facing adjacent and overlapping texts better than other representations; (2) the improvements of CPA and refined reference points are consistent for different text representations.

To further investigate the effect of the CPA module, Tab. 4 shows the performance of the models with different CPA configurations and the models replacing CPA with other fusion modules. The study of the CPA configuration has the following conclusions: (1) just enlarging features performs 0.8% worse; (2) only using attention has a minor 0.1% improvement; (3) combining both boosts the performance by 1.1% significantly. The effectiveness of the CPA module is that the attentional fusion adaptively highlights texts' features at a suitable scale and suppresses the features of other scales. Fig. 6 demonstrates the four attention maps across scales from shallow to deep layers of the backbone. Moreover, CPA performs 1.8% and 2.1% better than FPN and ASF (Liao et al., 2022). We attribute it to (1) FPN (Lin et al., 2017) will **NOT** improve the object detection performance because the cross-level feature exchange is already adopted by the multi-scale deformable attention module (Zhu et al., 2021a). FPN degrades text detection performance because introduced additional parameters are not learned well because the text dataset we used is much smaller than the common object dataset. (2) CPA's selective mechanism is more compatible with DETR-like detectors than ASF. The latter works better with DB++, which tends to preserve both the global structures and local details for accurate segmentation masks. A detailed comparison can be found at Sec. 6.2.

**Effect of shape-constrained loss.** Using the shape-constrained loss, the performance on Total-Text increases from 83.5% to 87.1%. As Fig. 5 shows, the model trained with shape-constrained loss can produce more complete contours (the right image in the pair) than the models without the loss (the left image in the pair).

## 5 CONCLUSION

We have presented PBFormer, an efficient and accurate text detection method. It is superior to handle crowded texts or texts with diverse shapes. PBFormer equips a new text representation, Polynomial Band, to a transformer-based network consisting of a cross-scale pixel attention module and a lightweight deformable transformer. We supervise the network with a shape-constrained loss term, encouraging the network to output the correct contour length. PBFormer shows strong robustness when training without pre-training on the additional datasets, which is much more resource-friendly than other transformer-based methods.

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

## 6 APPENDIX

♦ Sec. 6.1 states the differences between polynomial curve and Bezier curve.

♦ Sec. 6.2 states the differences between CPA and ASF (Liao et al., 2022).

♦ Sec. 6.3 states the architecture differences compared with TESTR (Zhang et al., 2022b) and Few-Better (Tang et al., 2022).

♦ Sec. 6.4 shows how we generate ground truth point sets for top, bottom, left and right sides from original annotated polygon points.

♦ Sec. 6.5 conducts experiments on DAST1500 (Tang et al., 2019), MLT2019 (Nayef et al., 2019), and ArT2019 (Chng et al., 2019) and show some qualitative results on them.

♦ Sec. 6.6 visualizes comparisons of different text representations with the same network.

♦ Sec. 6.7 investigate learning configurations of polynomial bands.

♦ Sec. 6.8 shows efficiency's comparisons of different text representations.

♦ Sec. 6.9 shows more visualizations on attention maps.

♦ Sec. 6.10 shows some failure cases.

♦ Sec. 6.11 shows more qualitative results on CTW1500 and Total-Text.

### 6.1 DIFFERENCE BETWEEN POLYNOMIAL CURVE AND BEZIER CURVE

The polynomial curve is another form of the fitting function, but a better form.

Let us consider the parameters optimized in our loss functions. For the polynomial curve, the optimized parameters are for the function defined in the image space. For the Bezier curve, the optimized parameters, *i.e.*, the control points, are for the function defined with a specific variable "$t$".

**Important**: The form of the curves will finally determine how we optimize their parameters. (1) For the polynomial curve, we can measure the difference between the annotated points and the predicted polynomial function by comparing evenly sampled indicated polynomial curve and ground truth text-line. Such a metric can better reflect how humans percept the shape difference. (2) For the Bezier curve, we need first to fit the ground truth to obtain the control points by least square method, then supervise the predicted Bezier curve by comparing the difference between the control points. Notably, the mismatch in Bezier's control point cannot reflect the difference in the curve shape. A slight difference in the control points may bring a large shape change.

**Can be Bezier curve learned like a polynomial curve in the image space?** The answer is **NO**. The Bezier curve is defined as:

$$c(t) = \sum_{j=0}^{n} b_j B_{j,n}(t), 0 \le t \le 1 \tag{13}$$

where $\{b_j | b_j = (x_j, y_j)\}_{j=0}^{n}$ are Bezier control points, and $B_{j,n}(t) = \begin{pmatrix} n \\ j \end{pmatrix} t^j (1-t)^{n-j}$ are

Bernstein basis polynomials. Please note that the Bezier curve $c(t)$ is polynomials of $t$ rather than $x$, with the value for $t$ being fixed between 0 and 1. If $c(t)$ samples $K+1$ points by taking $0, \frac{1}{K}, \frac{2}{K}, \ldots, 1$, the sampled points in the image space are not evenly arranged. Therefore, $c(t)$ cannot calculate loss with text-line in the image space.

For a clear understand, we give a simple Python code and visualization to show the above issue.

```python
#!/usr/bin/python3
#!--*-- coding: utf-8 --*--
import numpy as np
import matplotlib.pyplot as plt
from scipy.interpolate import interp1d
# Define Bezier control points
P0, P1, P2 = np.array([[0, 0], [2, 4], [5, 3]])
```

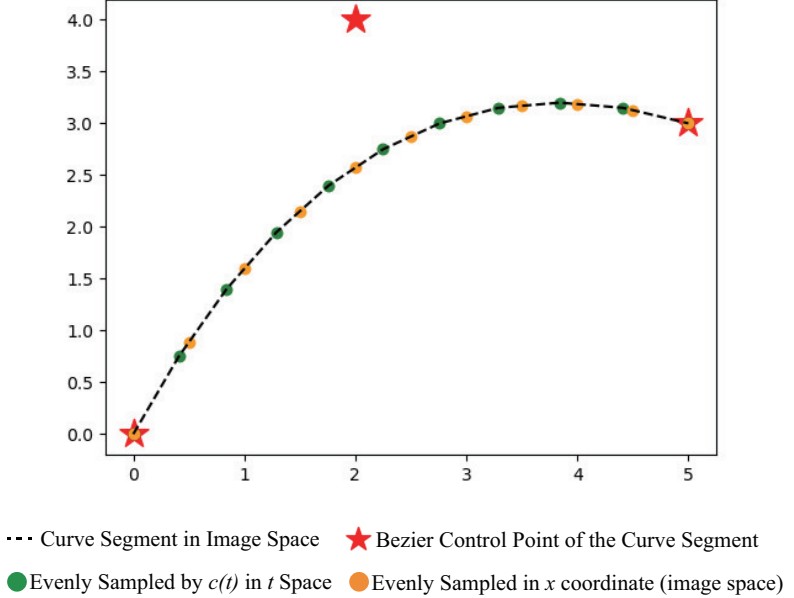

Curve Segment in Image Space ⋯⋯    ★ Bezier Control Point of the Curve Segment

● Evenly Sampled by $c(t)$ in $t$ Space    ● Evenly Sampled in $x$ coordinate (image space)

Figure 7: **The visualization of the mismatch about two point sequence.** Given a curve segment and its corresponding Bezier curve (represented by Bezier control points, red pentagram), points evenly sampled in image space are not matched with points evenly sampled in $t$ space of Bezier curve $c(t)$ in Eq. 13.

```
8  plt.scatter(P0[0], P0[1], c='red', s=300, marker='*')
9  plt.scatter(P1[0], P1[1], c='red', s=300, marker='*')
10 plt.scatter(P2[0], P2[1], c='red', s=300, marker='*')
11 # Define Bezier curve
12 P = lambda t: (1 - t)**2 * P0 + 2 * t * (1 - t) * P1 + t**2 * P2
13 # Equally sample 11 points in t space
14 points = np.array([P(t) for t in np.linspace(0, 1, 11)])
15 x, y = points[:, 0], points[:, 1]
16 plt.plot(x, y, color='black', linestyle='--')
17 plt.scatter(x, y, c='green')
18 # Equally sample 11 points in x space
19 y_fx = interp1d(x, y, fill_value="extrapolate")
20 _x = np.linspace(0, 5, 11)
21 _y = y_fx(_x)
22 plt.scatter(_x, _y, c='orange')
23 plt.show()
```

Listing 1: Python code to show evenly sampled points in Bezier's t space cannot be matched with evenly sampled points in image space.

As Fig. 7 shows, Given a curve segment (dashed line), we can sample 11 points from $x$=0 to $x$=5 with equally distance $\Delta_x$=0.5. Then, given a Bezier curve which represents the curve segment, we can also sample 11 points from $t$=0 to $t$=1 with equally distance $\Delta_t$=0.1. As we can see, sampled points evenly in $t$ space (green circles) are not matched with evenly sampled points in image space (orange circles). Thus Bezier curve cannot be learned like a polynomial curve in the image space.

## 6.2 DIFFERENCES BETWEEN CPA AND ASF.

Although CPA and ASP Liao et al. (2022) utilize multi-scale feature maps, they are very different in motivation and implementation.

Motivation difference:

- ASF works with DB++, a segmentation-based method to predict the pixel-wise text region. The motivation of ASF is to help DB++ obtain a more accurate segment mask, which includes both global structures and local details.

- CPA works with PBFormer, a detection-based method to predict the parameters of object shape. It is known that the detection results are sensitive to the scale, and the motivation of CPA is to highlight the best scale and suppress the others. Our experiments demonstrate that the selection mechanism of CPA is particularly crucial for DETR-like detectors (including our method) in text detection, which don't have non-maximum suppression(NMS) for post-processing. If we replace CPA with ASF, as in Table 4, the performance will drop 1.8% in terms of the F-score.

Implementation difference:

- ASF performs spatial attention. It first generates the mask weights for different feature maps by a stack of Conv-ReLU and Conv-Sigmoid modules, then concatenates the mask-weighted feature maps together.

- CPA performs scale attention. It aligns the feature maps of the different scales by enlarging the small-scale feature maps. The attention map is not obtained by introducing a new network module but by comparing the values of the existing feature maps among different scales with SoftMax. All the feature maps are weighted with the attention map the resized to the original size.

Explanation summary:

- To preserve both the global structure and local details of the segmentation mask, ASF performs spatial attention for each feature map and concatenates the information of all the feature maps.

- To produce the detection results with an accurate scale, CPA performs scale attention among different feature maps and selectively highlights the feature maps of the best scale and suppresses the feature maps of the other scale.

- We assume the selective mechanism of CPA is compatible with DETR-like detectors that do not have non-maximum suppression. Two pieces of evidence: if we replace CPA with ASF, the performance drop; We only utilize two-layer transformer encoders and decoders, much less than the TESTR, indicating the effectiveness of the fused feature map.

### 6.3 RELATIONS TO TESTR AND FEWBETTER

PBFormer has multiple efficient and useful designs compared with TESTR (Zhang et al., 2022b) and FewBetter (Tang et al., 2022).

Compared with TESTR:

- PBFormer is single-stage without relying on intermediate bounding box results. In contrast, TESTR adopts the two-stage Deformable-DETR. It predicts the bounding box in each feature point of the transformer encoder's output, then selects topK boxes based on confidence to embed them into positional embeddings and reference points, as Fig. 8 (a)'s red block shows.

- PBFormer adopts a coarse-to-fine strategy to generate reference points for deformable attention modules. The first decoder's output encodes 2-d residuals to move the reference point into a better localization. In contrast, TESTR uses the same reference points for all deformable attention modules. Fig. 8 (a) and (c)'s orange arrows illustrate the differences.

- PBFormer inserts a Cross-scale Pixel Attention module, which selectively highlight text regions at suitable scales. The selective mechanism of CPA is compatible with DETR-like detectors that do not have non-maximum suppression, enabling PBFormer to only utilize two layers of transformer encoders and decoders. In comparison, TESTR uses six layers of transformer encoders and decoders.

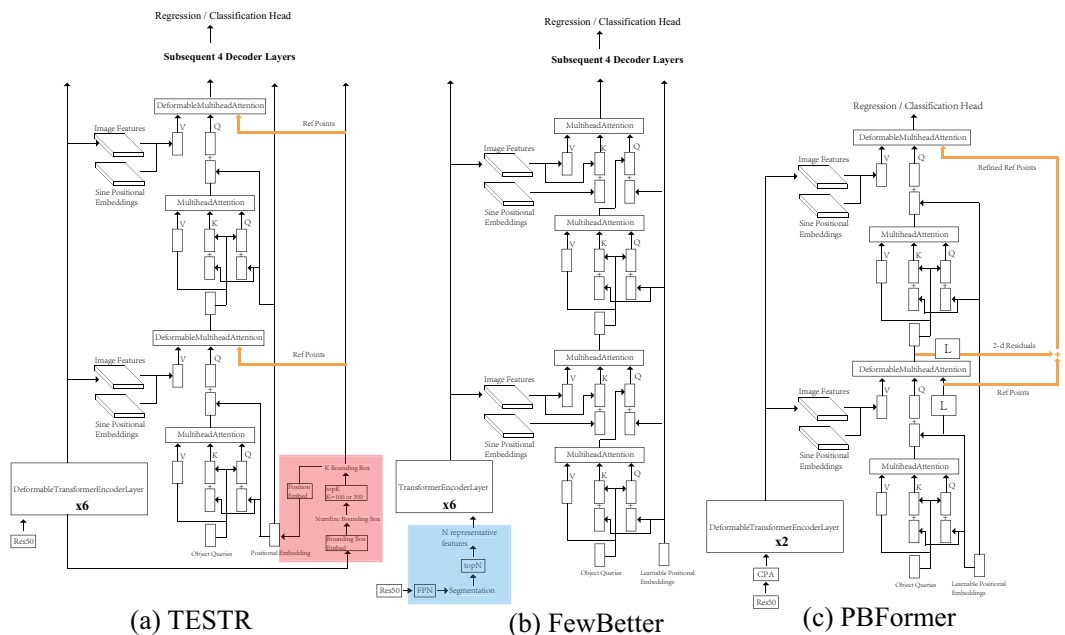

(a) TESTR          (b) FewBetter          (c) PBFormer

Figure 8: **Detailed Network Comparisons with TESTR and FewBetter.** (a) NumEnc denotes the sequence length of encoder's output, which equals the total spatial quantity across multi-scale feature maps. (c) The proposed CPA module has not any trainable parameters.

- PBFormer is only pre-trained on CurvedSynthText, while TESTR is pre-trained on a combination of three datasets, *i.e.*, CurvedSynthText, MLT2017, and Total-Text.

Compared with FewBetter:

- PBFormer does not need FPN in the backbone or generate segmentation maps. Differently, FewBetter has an FPN network to generate convincing segmentation masks for feature selections, as Fig. 8 (b)'s blue block shows.
- PBformer utilizes a deformable transformer while FewBetter uses the original transformer. Besides, we also use two transformer encoder and decoder layers, while FewBetter uses the six for encoder and decoder, respectively.

## 6.4 GROUND TRUTH OF TOP, BOTTOM, LEFT AND RIGHT GENERATION.

In this section, we introduce how to generate four ground truth point-sets for the top, bottom, left and right sides.

In common text datasets, a text instance is annotated by $2K$ discrete points. For example, $K = 5$ in Total-Text and $K = 7$ in CTW1500. More importantly, these points are annotated along with human reading hobbies. They are ordered in counterclockwise order, and the first point is always the top-left of the first character. The former $K$ points $\mathbf{p}_1, ...\mathbf{p}_K$ and the latter $K$ points $\mathbf{p}_{K+1}, ...\mathbf{p}_{2K}$ are located as Fig. 9 shows (green for the former $K$, blue for the latter $K$).

Therefore, the generating process of ground truths for the top, bottom, left, and right curve follows subsequent five steps:

1. Divide $2K$ annotations into four Sets. $\mathbf{S}^1 = \{\mathbf{p}_1, \ldots, \mathbf{p}_K\}$, $\mathbf{S}^2 = \{\mathbf{p}_{K+1}, \ldots, \mathbf{p}_{2K}\}$, $\mathbf{S}^3 = \{\mathbf{p}_K, \mathbf{p}_{K+1}\}$, and $\mathbf{S}^4 = \{\mathbf{p}_{2K}, \mathbf{p}_1\}$.
2. Assuming $\mathbf{S}^1$ and $\mathbf{S}^2$ would be fitted by a mapping $f : x \rightarrow y$, judge whether assumed functions are both single-valued. If yes, go to step 3. If not, go to step 4.
3. $\mathbf{S}^1$ and $\mathbf{S}^2$ both use the form $y = f(x)$; $\mathbf{S}^3$ and $\mathbf{S}^4$ both use the form $x = f(y)$. If $\mathbf{S}^1$'s middle point is higher than $\mathbf{S}^2$'s, $\mathbf{S}^1$ becomes the ground truth for the top curve, and $\mathbf{S}^2$

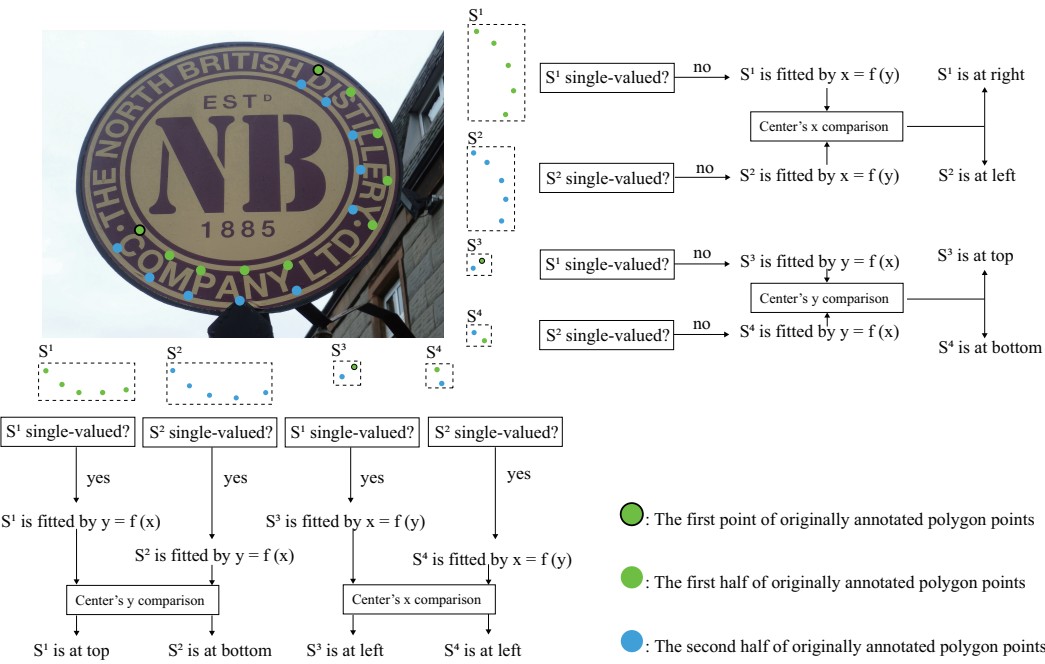

Figure 9: **Visualization of generating ground truth for the top, bottom, left, and right sides from raw annotations.** The black rounded green dots represent the first annotated point for text.

    becomes the ground truth for the bottom curve, and vice versa. If $\mathbf{S}^3$'s middle point is more left than $\mathbf{S}^4$'s, $\mathbf{S}^3$ becomes the ground truth for the left curve, and $\mathbf{S}^4$ becomes the ground truth for the right curve, and vice versa. Go to step 5.

4. $\mathbf{S}^1$ and $\mathbf{S}^2$ both use the form $x = f(y)$; $\mathbf{S}^3$ and $\mathbf{S}^4$ both use the form $y = f(x)$. If $\mathbf{S}^1$'s middle point is more left than $\mathbf{S}^2$'s, $\mathbf{S}^1$ becomes the ground truth for the left curve, and $\mathbf{S}^2$ becomes the ground truth for the right curve, and vice versa. If $\mathbf{S}^3$'s middle point is higher than $\mathbf{S}^4$'s, $\mathbf{S}^3$ becomes the ground truth for the top curve, and $\mathbf{S}^4$ becomes the ground truth for the bottom curve, and vice versa. Go to step 5.

5. Sample $\mathbf{S}^1$, $\mathbf{S}^2$, $\mathbf{S}^3$, and $\mathbf{S}^4$ evenly to generate dense supervision points (for PB's loss calculation, see Sec. 3.3).

**Single-valued condition.** A mapping $f : x \rightarrow y$ which is used to fit a set of order points satisfies the single-valued condition if and only if:

$$x_1 < x_2 < ... < x_n \vee x_1 > x_2 > ... > x_n, \tag{14}$$

where $x_1, \ldots, x_n$ is the x-coordinate sequence of order points, and $n$ denotes the number of points.

As Fig. 9 shows, we illustrate the processes for a horizontal reading direction text "COMPANY" and a vertical reading direction text "DISTILLERY."

Table 5: Comparisons on DAST1500's testing set.

| Method | F. | Prec. | Rec. |
|---|---|---|---|
| TextBoxes | 50.9 | 67.3 | 40.9 |
| RRD | 53.0 | 67.2 | 43.8 |
| EAST | 62.0 | 70.0 | 55.7 |
| SegLink | 65.3 | 66.0 | 64.7 |
| CTD+TLOC | 66.6 | 73.8 | 60.8 |
| PixelLink | 74.7 | 74.5 | 75.0 |
| ICG | 79.4 | 79.6 | 79.2 |
| ReLaText | 85.8 | 89.0 | 82.9 |
| MAYOR | **86.6** | 87.8 | **85.5** |
| PBFormer(ours) | 85.9 | 90.0 | 82.1 |
| PBFormer(two-stage) | **86.6** | **90.2** | 83.2 |

Table 6: Experiments on MLT2019 and Art2019.

| Method | Rep. | F.MLT | F.Art |
|---|---|---|---|
| DB++ | Seg | 64.7 | 71.1 |
| FCENet | Seg | 66.3 | 74.0 |
| TESTR | Pts | 68.7 | 75.2 |
| TESTR | Bez | 66.7 | 73.6 |
| PBFormer(ours) | PB | **69.3** | **76.2** |

DAST1500

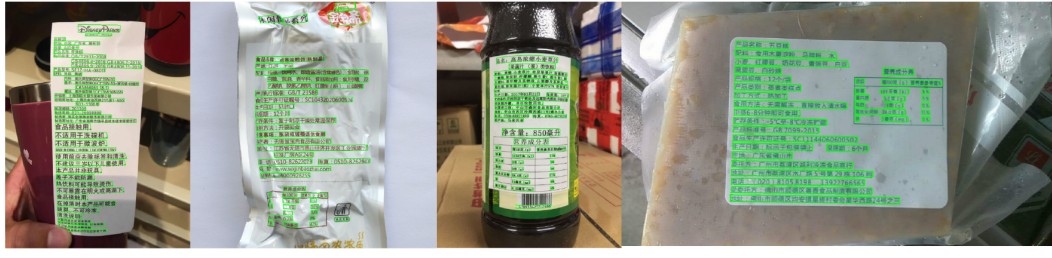

Art2019

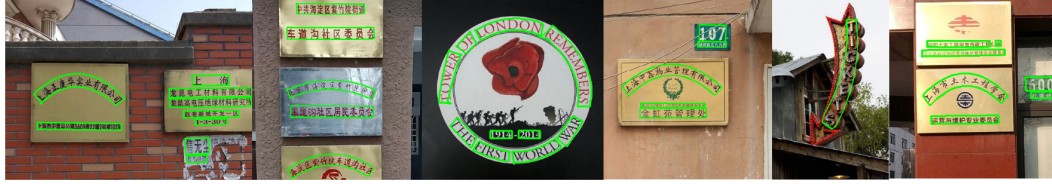

MLT2019

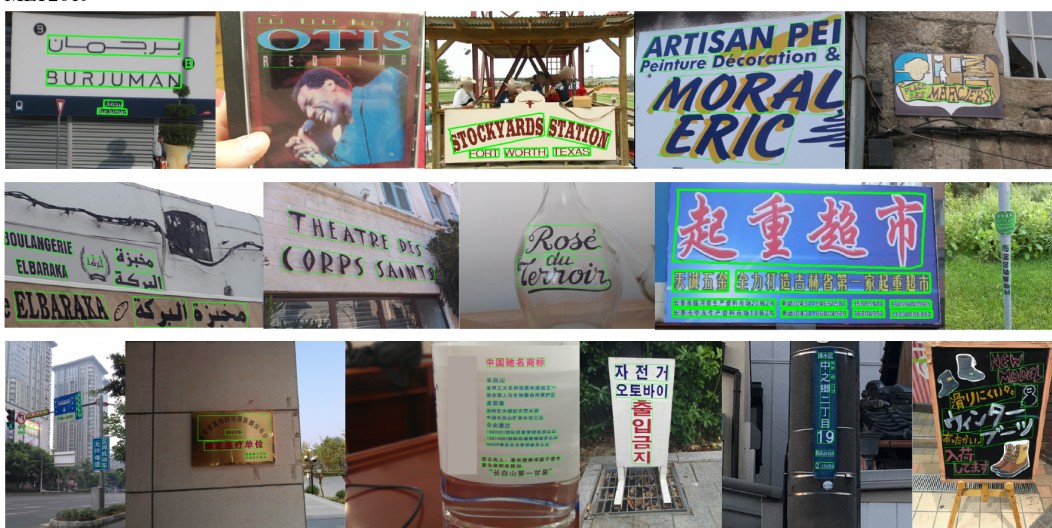

Figure 10: **Qualitative results on DAST1500, ArT2019, and MLT2019.**

## 6.5 EXPERIMENTS ON DAST1500, MLT2019, AND ART2019

We follow MAYOR (Qin et al., 2021) to conduct experiments on DAST1500 (Tang et al., 2019) and compare with TextBoxes (Liao et al., 2018a), RRD (Liao et al., 2018b), EAST (Zhou et al., 2017), SegLink (Shi et al., 2017), CTD+TLOC (Liu et al., 2019), PixelLink (Deng et al., 2018), ICG (Tang et al., 2019), ReLaText (Ma et al., 2021), MAYOR (Qin et al., 2021).

Methods on leaderboards of ICDAR MLT 2019 (Nayef et al., 2019) and ICDAR ArT 2019 (Chng et al., 2019) adopt ResNeSt-101 (Zhang et al., 2022a), ResNeXt-101 (Xie et al., 2017), or ResNext-152 (Xie et al., 2017) as their backbones which are far heavy for a fair comparison. Therefore, we conduct experiments across DB++ (Liao et al., 2022), FCENet (Zhu et al., 2021b), TESTR (Zhang et al., 2022b), and PBFormer by splitting the training set of MLT2019 and Art2019 into training and validation with 1:1. We generate ground truth of Bezier control points following ABCNet (Liu et al., 2020). We fine-tune all models' best checkpoints on the training split 1000 epoch.

Results are shown in Tab. 5 and Tab. 6. PBFormer achieves comparable results with previous state-of-the-art on DAST1500. We attribute to (1) DAST1500 are full of dense texts. MAYOR is two-stage and has spatial designs for those cases, such as region proposal network (RPN) and MLP Mask Decoder (MLM). (2) PBformer is single-stage. Although it performs less well than MAYOR, it is

more efficient because it does not have post-processing with NMS. If we take training strategy same with two-stage Deformable-DETR, we can achieve better performance (86.7 vs. 86.6). Besides, one-stage PBFormer still outperforms MAYOR on CTW1500 (87.0 vs. 85.3) and Total-Text (87.1 vs. 86.3).

Moreover, PBFormer outperforms DB++, FCENet, and TESTR on MLT 2019 and Art 2019. We also show some visualizations on DAST1500, MLT2019, and ArT2019 in Fig. 10.

### 6.6 VISUALIZED COMPARISONS OF DIFFERENT TEXT REPRESENTATIONS WITH THE SAME NETWORK.

In this section, we adopt the same network architecture, PBFormer, but change the output head using Bezier control points, polygon points, and polynomial bands. In Fig. 12, we show comparisons of adjacent and overlapping texts. For adjacent texts, PB can distinguish individual texts correctly, but the others are likely to combine them falsely. For overlapping texts, PB can detect complete contours, while others are likely to detect incomplete contours or lose the overlapped texts.

### 6.7 INVESTIGATION OF PB'S CONFIGURATIONS.

Table 7: Comparisons of the number of different supervision points per polynomial curve.

| Number | F. | Prec. | Rec. |
|--------|------|-------|------|
| 6 | 84.6 | 90.5 | 79.5 |
| 12 | 86.0 | 90.4 | 81.9 |
| 24 | **87.1** | **92.1** | **82.6** |
| 30 | 86.5 | 91.1 | 82.4 |
| 36 | 86.4 | 91.1 | 82.3 |

Table 8: Comparisons of polynomial's order.

| Curve | F. | Prec. | Rec. |
|-----------|----------|----------|----------|
| Quadratic | **87.1** | 92.1 | 82.6 |
| Cubic | 87.0 | **92.2** | 82.4 |
| Quartic | 86.7 | 91.0 | **82.7** |

We first analyze the influence of the polynomial order of PB. Theoretically, higher-order polynomials could fit more complex texts, but the quadric curves can describe the texts in most current datasets well. For this reason, the higher-order curves are overqualified to represent the text shapes in the existing datasets, but they are easily overfitted. Secondly, we analyze the effect of sampling points' density. As Tab. 7 shows, too few points perform 2.5% worse since the quantity is insufficient to learn curved shapes, while too many points also decrease 0.7% due to redundant learning points, especially for straight texts.

### 6.8 EFFICIENCY'S COMPARISONS OF DIFFERENT TEXT REPRESENTATION

The polynomial band is more efficient than segmentation masks and polygon points during inference, as Fig. 11 shows. The polynomial band is more efficient than Bezier points during ground truth generations.

Polygon points need spline interpolation because the number of network's output is too sparse for evaluation, such as 16 points in TESTR, while more points are needed to compute accurate IoUs in MMOCR's evaluation protocol (Kuang et al., 2021), especially for curved texts. Therefore, as Fig. 11 shows, after dividing points into splines, each of which is fitted by a polynomial, then evenly sampling. In contrast, PB avoids the above two-stage iterative procedure because PB direct contains four curves for the whole contour, which generates a dense contour by sampling.

Ground truths of Bezier points are calculated by the **least square algorithm** based on original polygonal annotations (Liu et al., 2020), which consumes additional time and resources for generating ground truth for extra data. Differently, PB directly utilizes the polygonal annotations without additional consumption to transform raw annotations.

### 6.9 MORE VISUALIZATIONS ON ATTENTION MAPS.

In Fig. 13, we select some representative attention maps of the cross-scale pixel attention module. The CPA is capable of attending to text regions at a suitable layer adaptively. We see CPA learned to attend at a single layer if texts have similar sizes. When different texts have size various, CPA learned to attend to relatively small texts at the swallow layer while large texts at the deep layer.

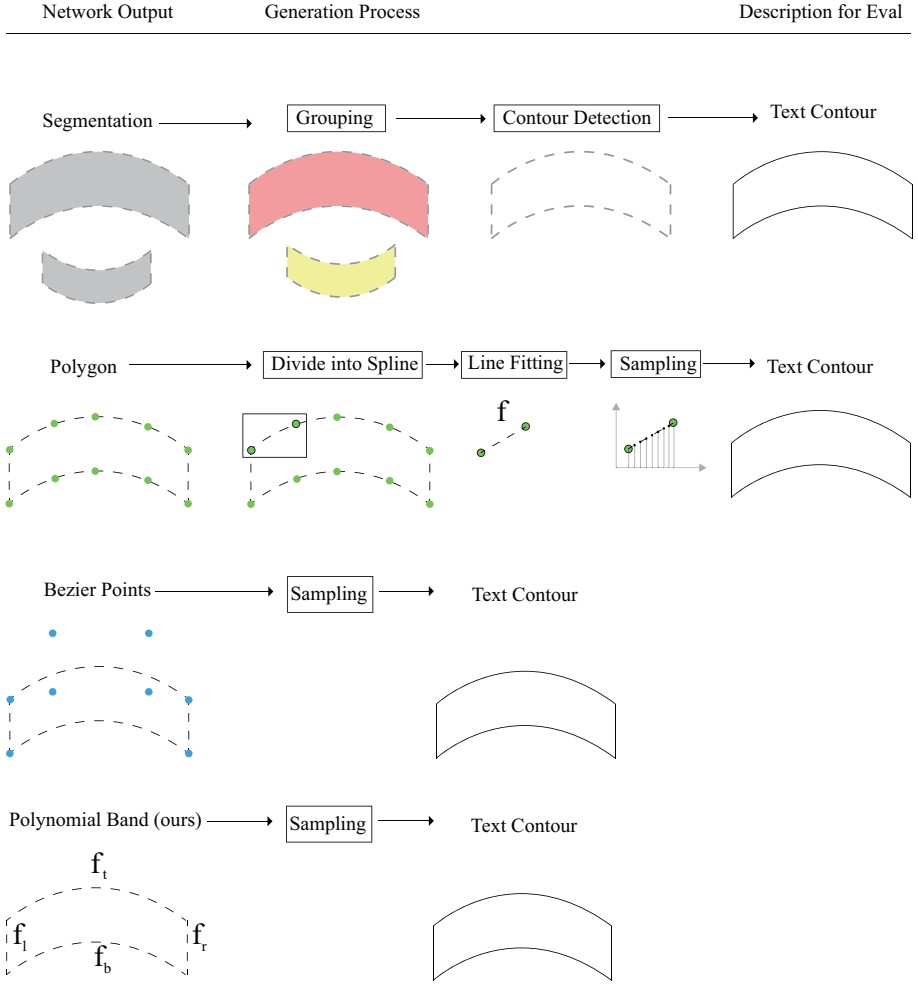

Figure 11: **Comparisons of the process from network's output representation to final text contour.** For segmentation maps, we only show main post-processing since there might be other practical processes such as outlier removal, distance (or directional) maps refinement, etc.

### 6.10 FAILURE CASES.

In Fig. 14, we visualize failure cases in two situations: (1) texts which are extremely small as Fig. 14(a) shows; (2) texts which occupy a minority in common datasets, such as chinese-lingual texts in Fig. 14(b) and texts processed into symbols or beautified by art-style.

The limitations may inspire our future work. Firstly, enlarging the input's resolution can improve the detection of small texts, but it further consumes more computations. Secondly, collecting multilingual texts or adopting art-style augmentation on the common texts for training will improve the recall of the results.

### 6.11 MORE QUALITATIVE RESULTS.

In Fig. 15 and Fig. 16, we present more qualitative results on CTW1500 and Total-Text. PBFormer predicts not only accurate contour for curved texts but also handles multi-oriented data successfully. More importantly, when texts are crowded, PBFormer also distinguishes them clearly. Results indicate that our PBFormer is robust in various scenes.

Bezier Control Points      Polygon Points      Polynomial Band

Adjacent

Overlapping

Figure 12: **Visualized comparisons of different text representations with the same network.**

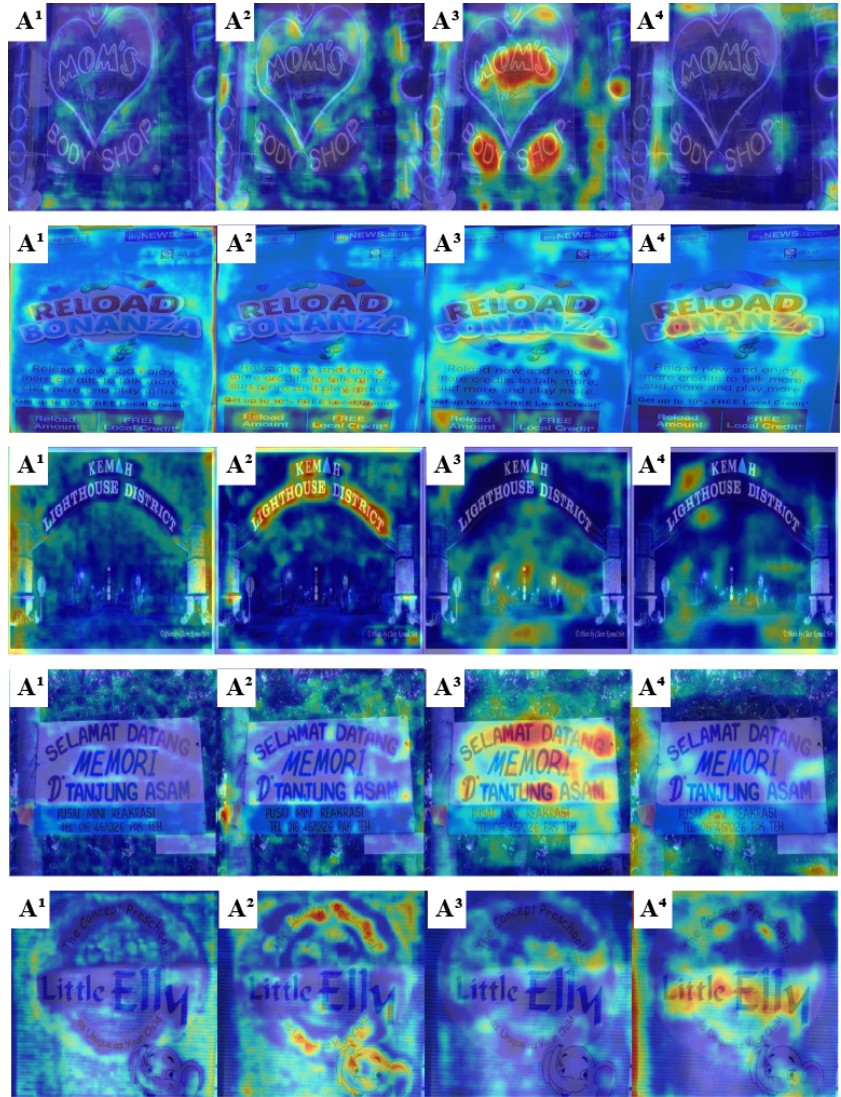

Figure 13: **CPA's attention maps on Total-Text.** $A^1$, $A^2$, $A^3$ and $A^4$ represent the attention map weighting the multi-scale feature from network's swallow to deep layers.

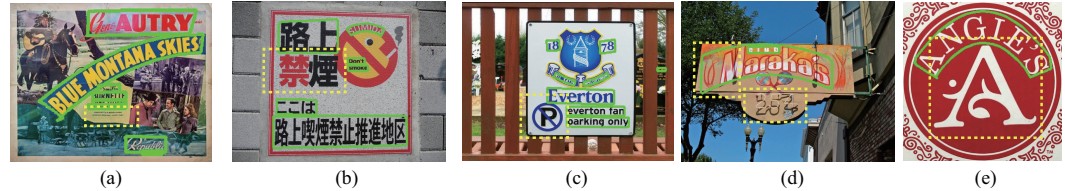

Figure 14: **Visualization of failure cases.** The yellow dashed boxes denote the failed detections.

Curved Texts

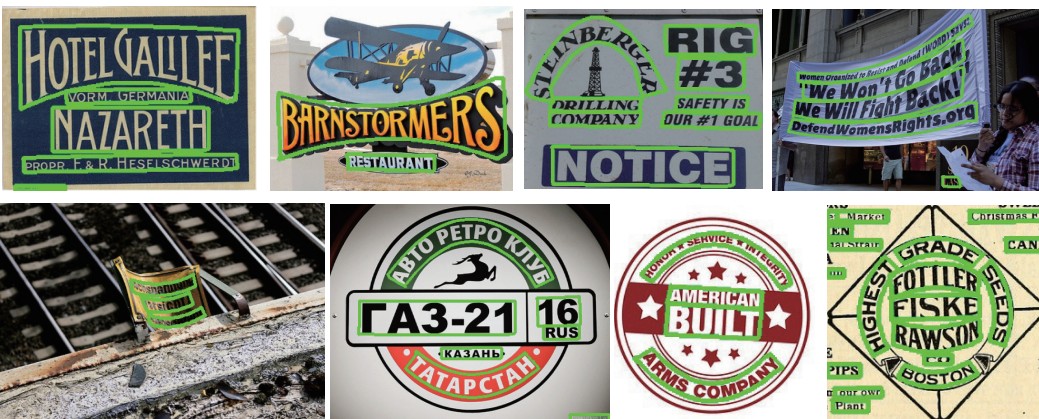

Crowded Texts

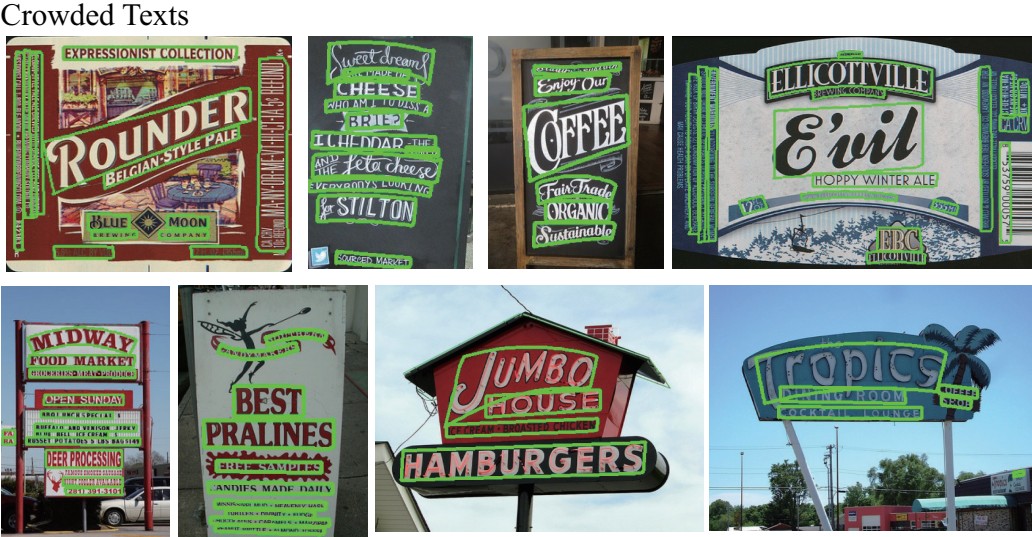

Multi-oriented Texts

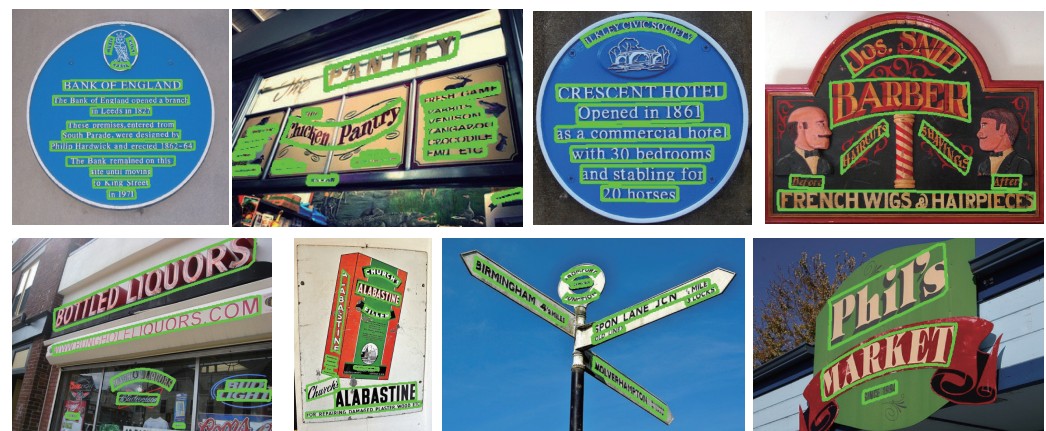

Figure 15: **More Qualitative results on CTW1500.** We demonstrate different types of texts, such as curved, crowded, or multi-oriented.

Curved Texts

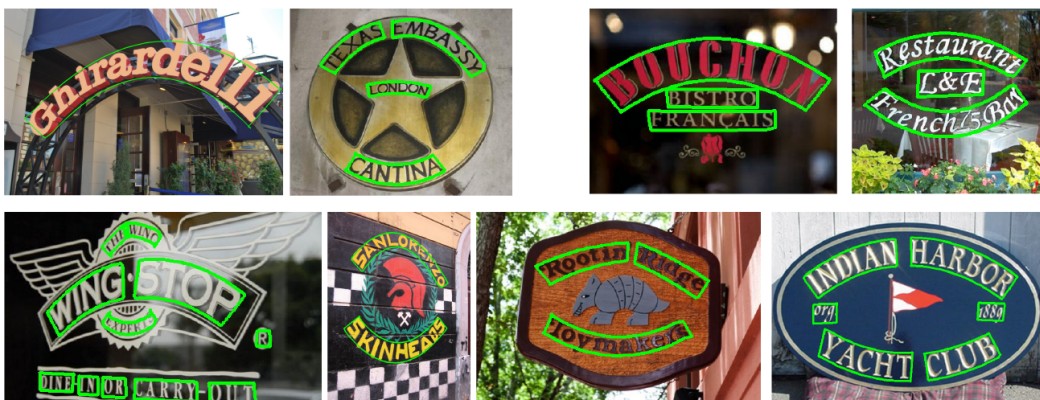

Crowded Texts

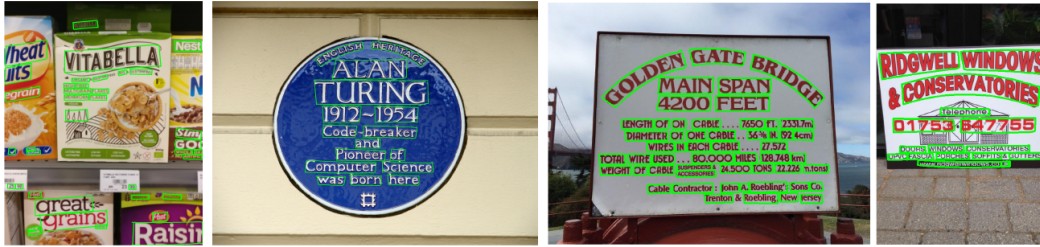

Multi-oriented Texts

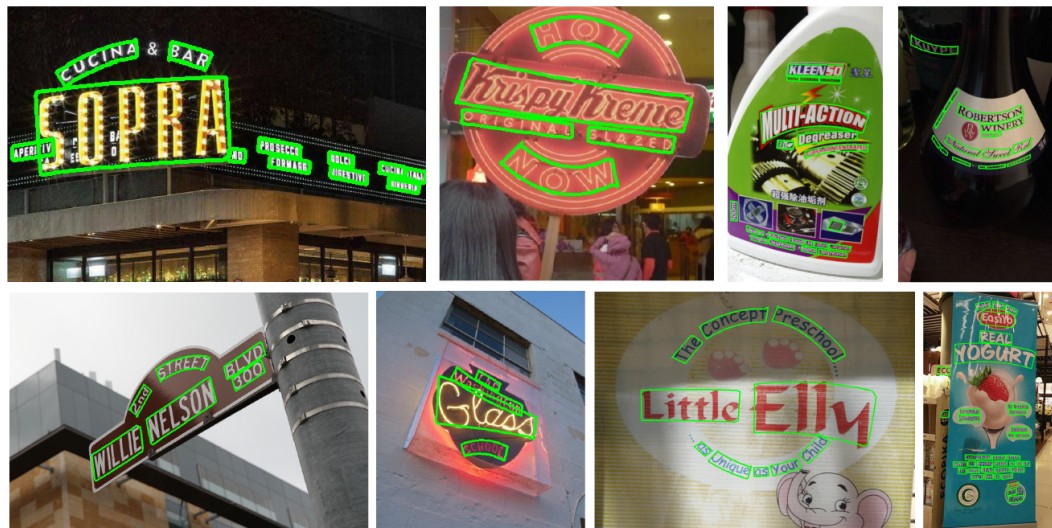

Figure 16: **More Qualitative results on Total-Text.** We demonstrate different types of texts, such as curved, crowded, or multi-oriented.

