# OpenReview forum: "PBFormer: Capturing Complex Scene Text Shape with Polynomial Band Transformer"
_ICLR.cc/2023/Conference — Submitted to ICLR 2023_

### Official Review · Reviewer_FmtQ · 2022-10-21

**Confidence:** 4
**Correctness:** 3
**Technical Novelty And Significance:** 2
**Empirical Novelty And Significance:** 2
**Recommendation:** 3

**Clarity, Quality, Novelty And Reproducibility:**

I would suggest to use some equations to explain lightweight deformable transformer, but overall, the paper is easy to follow. The proposed text representation has some novelty and it could have a good impact within the community as the absolute numbers look great.

**Strength And Weaknesses:**

Strengths:

A new text representation, PB, is a concise and powerful representation for most texts observed in wild. The shape-constrained loss is well designed for the representation.

CPA is parameter-free and computationally inexpensive, but it is shown that it is better than other methods.

The reported absolute numbers look strong.

Weaknesses:

In section 3.2, it discusses a potential weakness of the representation that it cannot represent lines that have multiple values for single x. Probably, it is true that it is rare to observe such cases in wild, but it is a fundamental limitation of the approach.

Text detection is often followed by text recognition using the detected boxes. Typically, features corresponding to the bounding boxes are extracted and fed to the recognition model. The paper does not discuss how to extract features (or pixels) if a text is represented by PB.

It was not clear how a text is represented by four lines. I guess there is no guarantee that the beginning and ending points of each line match each other and there should be gaps among the lines. In that case, how is a complete region defined given four lines (and how is IoU computed)?

It was not clear if the proposed representation was good or the model was good. Ideally, other representations should be implemented with the same model and perform fair comparisons among them.

The results in Table 3 are interesting. Assuming "-" is the baseline without anything, adding something other than CPA did not seem to help much or even made it worse. I was not sure if it was reasonable. If there was a good explanation on this, it would be good to have a discussion on it.


**Summary Of The Paper:**

This paper presents a new representation of texts named Polynomial Band (PB) for text detection. The shape-constrained loss designed for PB considers both shape and the length. The model uses ResNet50 as the backbone followed by Cross-scale Pixel Attention (CPA) and Lightweight Deformable Transformer. CPA is parameter-free, but it can effectively suppress useless regions and thus improves the final results. The lightweight deformable transformer helps to achieve high FPS values. The experimental results show the effectiveness of the approach.

**Summary Of The Review:**

The results are good and the representation seems powerful and reasonable. Other proposals are also effective. Therefore, I see a good value in the paper for the scene text recognition community. However, there are several weaknesses found in the paper as mentioned above and I am not fully convinced that this paper should be accepted in the current form.

---

> ### Author Response · Authors · 2022-11-19
> **2 The way to extract features (or pixels) if a text is represented by PB**
>
> We can obtain features (or pixels) inside a text contour by Hard RoI masking, which is proposed by Mask TextSpotter v3.
>
> (1) Firstly, it uses the minimum, axis-aligned, rectangular *bounding boxes* of the **polygon proposals** to generate RoI features by RoI Align.
>
> (2) Secondly, it multiplies *binary polygon masks* with the RoI features to suppress background noise or neighboring text instances, where a polygon mask M indicates an axis-aligned rectangular binary map with all 1 values in the polygon region and all 0 values outside the polygon region. M can be easily generated by filling the **polygon proposal** region with 1 while setting the values outside the polygon to 0.
>
>
> For PB, it can generate a polygon by sampling the top curve from left to right, sampling the right curve from top to bottom, sampling the bottom curve from right to left, sampling the left curve from bottom to top, then concatenating them. Then
>
> (1) The **polygon generated from PB** can build a *bounding box* by finding minimum and maximum values across x and y coordinates, then generate RoI features as the above (1) states.
>
> (2) The **polygon generated from PB** can also make *binary polygon masks*, following the above (2) steps.
>
> Thus, if a text is represented by PB, features (or pixels) of a text can be extracted following Hard RoI masking of Mask TextSpotter v3.
>
> Mask TextSpotter v3: Segmentation Proposal Network for Robust Scene Text Spotting. [ECCV'2020]

---

> > ### Comment · Reviewer_FmtQ · 2022-11-25
> > **Can this be used only with attention-based models?**
> >
> > Thank you for the explanation. It would be great to put these details in the paper.
> >
> > It sounds like this might not work the best with recognition methods with CTC (or left-to-right methods in general), that is still commonly used in practical systems, because it does not rectify the lines. For scene text benchmarks, transformer-based decoder (or attention-based seq2seq) might be the current best, but it would be better if it works well with other methods, too.

---

> > > ### Author Response · Authors · 2022-11-30
> > > **The way to use our method with recognition methods with CTC (or left-to-right methods in general)**
> > >
> > > Our method can also be used with recognition methods with CTC (or left-to-right methods in general) by a rectifying module following the GTC method.
> > >
> > > (1) GTC uses an STN to transform input images into rectified images. As many text images in natural scenes appear with curved texts and different perspectives, the transformation module is adopted for robust and accurate recognition. It is a differentiable module.
> > >
> > > (2) After getting the rectified image, GTC applies its network and supervises it using the CTC loss.
> > >
> > > For our method,
> > >
> > > (1) we can also insert an STN to transform an extracted image patch into a rectified image. To extract an image patch, the PB can generate a polygon and then build a bounding box for RoI cropping by finding minimum and maximum values across x and y coordinates.
> > >
> > > (2) after getting the rectified image, we can use CTC loss for text recognition following GTC.
> > >
> > > Spatial Transformer Networks[NIPS'2015]
> > >
> > > GTC: Guided Training of CTC Towards Efficient and Accurate Scene Text Recognition. [AAAI'2020]

---

> ### Author Response · Authors · 2022-11-19
> **3 Responses to "How a text is represented by four lines?", gaps between adjacent lines, "How is a region defined given four lines" and "How is IoU computed"**
>
> **How is a text represented by four lines?**
>
> Commonly, a text is represented by a contour which is a point set. Point order is usually arranged in a counterclockwise sequence.
>
> In our method, we split the point set into four subsets, one for the top side, one for the bottom, one for the left, and one for the right.
> PBFormer learns to output four lines, each of which fits one of the above four subsets.
>
> Finally, four lines of PB still represent a text by building a contour. Points are sampled by the top line from left to right, the right line from top to bottom, the bottom line from right to left, and the left line from bottom to top, then concatenating, which is still arranged in a counterclockwise sequence.
>
> **Guarantees about gaps**
>
> There are no gaps among the four sides' GT point sets.
>
> There is a guarantee that the beginning and ending points of each side match each other. Please see step 1 of generating process in Appendix Section 6.4.
>
> The first point in $S^3$ is the same as the last of $S^1$,
> and the last point in $S^3$ is the same as the first of $S^2$.
> The first point in $S^4$ is the same as the last of $S^2$,
> and the last point in $S^4$ is the same as the first of $S^1$.
> After assigning them to four sides, those guarantees:
>
> (1) the top's rightmost GT is the same as the right's topmost GT;
>
> (2) the right's bottommost GT is the same as the bottom's rightmost GT;
>
> (3) the bottom's leftmost GT is the same as the left's bottommost GT;
>
> (4) the left's topmost GT is the same as the top's leftmost GT.
>
> With the above GT's guarantee, gaps between predicted adjacent lines are very close so that we can directly connect a line's last point with the next line's first point.
>
>
> **How is a complete region defined given four lines?**
>
> A complete region is represented by a contour which is a point sequence with a counterclockwise order. Given four lines, we can construct a contour by sampling the top curve from left to right, sampling the right curve from top to bottom, sampling the bottom curve from right to left, sampling the left curve from bottom to top, then concatenating those sampled points, which is still a point sequence with a counterclockwise order.
>
>
> **How is IoU computed?**
>
> The IoU is computed between a GT contour and a PRED contour based on Python's *shapely.geometry.Polygon* toolkit (in MMOCR's evaluation protocol). The GT contour itself is a sequence of points arranged counterclockwise. The PRED contour can also be generated from the four lines of PB, as we stated above.

---

> ### Author Response · Authors · 2022-11-19
> **4 Fair comparisons about the same model with different representations**
>
> We conduct the experiments with the same architecture (the same as ours ) but different shape representations.
> This Table presents ablations for various representation choices. With the same network, the performance of the polynomial band (PB) always outperforms polygon points or Bezier curves. The visualization results in appendix Appendix Section 6.6 also show that the PB can produce more accurate shapes of "distinguishing adjacent or overlapping texts" than other representations under the same network. All the reported results are also displayed in Section 4.2 and visualized in Appendix Section 6.6.
>
>
> | Method | CPA | Refine-Ref. | Rep. |F. | Prec. | Rec. |
> | :-----: | :-----: | :---------: | :-----: |:-----: | :-----: | :-----: |
> | PBFormer | - | - | PB | **85.8** | **90.2** | **81.9** |
> | PBFormer | - | - | Pts | 82.8 | 88.4 | 77.8 |
> | PBFormer | - | - | Bez | 83.9 | 89.8 | 78.7 |
> | PBFormer | - | &check; | PB | **86.0** | **90.5** | **81.9** |
> | PBFormer | - | &check; | Pts | 83.2 | 87.6 | 79.1 |
> | PBFormer | - | &check; | Bez | 84.2 | 89.7 | 79.4 |
> | PBFormer | &check; | &check; | PB | **87.1** | **92.1** | **82.6** |
> | PBFormer | &check; | &check; | Pts | 84.6 | 90.5 | 79.5 |
> | PBFormer | &check; | &check; | Bez | 85.3 | 90.9 | 80.4 |
>
>
>
>
> To further validate the effect of PB, we also conducted experiments when comparing PBFormer-- and TESTR. Here PBFormer-- means PBFormer without proposed CPA & refined References. This Table shows that using TESTR's network which uses two-stage Deformable-DETR and more transformer layers, PB also outperforms other output heads.
> | Method     | Basic Model  | #TR        | Rep.   | F.      | Prec.   | Rec. |
> | :-----    | :-----      | :---:      | :-----: |:-----: | :-----: | :-----: |
> | PBFormer-- |Deformable-DETR            | 2      | Pts  | 82.8 | 88.4 | 77.8 |
> | PBFormer-- |Deformable-DETR            | 2      | Bez  | 83.9 | 89.8 | 78.7 |
> | PBFormer-- | Deformable-DETR           | 2      | PB   | **85.8** | **90.2** | **81.9** |
> | TESTR      | two-stage Deformable-DETR | 6      | Pts  | 85.3 | 89.7 | 81.2 |
> | TESTR      | two-stage Deformable-DETR | 6      | Bez  | 86.3 | 90.3 | 82.6 |
> | PBFormer-- | two-stage Deformable-DETR | 6      | PB   | **87.4** | **91.9** | **83.3** |

---

> ### Author Response · Authors · 2022-11-19
> **5 Explanations about CPA's effectiveness compared with other methods**
>
> **Why does FPN not work?**
> FPN will not improve the object detection performance because the cross-level feature exchange is already adopted by the multi-scale deformable attention module. This has been verified by  Deformable-DETR (Sec. 5.2 and Tab. 2 in their paper). FPN degrades text detection performance because introduced additional parameters are not learned well due to the text dataset we used is much smaller than the common object dataset (1k-1.2k  vs. 118k).
>
>
> **Why is CPA better than ASF?**
> Although CPA and ASF utilize multi-scale feature maps, they are very different in motivation and implementation.
>
>
> **Motivation Difference:**
>
> (1) ASF works with DB++, a segmentation-based method to predict the pixel-wise text region. The motivation of ASF is to help DB++ obtain a more accurate segment mask, which includes both global structures and local details.
>
> (2) CPA works with PBFormer, a detection-based method to predict the parameters of object shape. It is known that the detection results are sensitive to the scale, and the motivation of CPA is to highlight the best scale and suppress the others. Our experiments demonstrate that the selection mechanism of CPA is particularly crucial for DETR-like detectors (including our method) in text detection, which don't have non-maximum suppression(NMS) for post-processing. If we replace CPA with ASF, as in Table 3, the performance will drop 1.8% in terms of the F-score.
>
>
> **Implementation Difference:**
>
> (1) ASF performs spatial attention. It first generates the mask weights for different feature maps by a stack of Conv-ReLU and Conv-Sigmoid modules, then concatenates the mask-weighted feature maps together.
>
> (2) CPA performs scale attention. It aligns the feature maps of the different scales by enlarging the small-scale feature maps. The attention map is not obtained by introducing a new network module but by comparing the values of the existing feature maps among different scales with softmax. All the feature maps are weighted with the attention map the resized to the original size.
>
>
> **Explanation Summary:**
>
> (1) To preserve both the global structure and local details of the segmentation mask, ASF performs spatial attention for each feature map and concatenates the information of all the feature maps.
>
> (2) To produce the detection results with an accurate scale, CPA performs scale attention among different feature maps and selectively highlights the feature maps of the best scale, and suppresses the feature maps of the other scale.
>
> (3) We assume the selective mechanism of CPA is compatible with DETR-like detectors that do not have non-maximum suppression. Two pieces of evidence: if we replace CPA with ASF, the performance drop; We only utilize two-layer transformer encoders and decoders, much less than the TESTR, indicating the effectiveness of the fused feature map.
>
>
> **Why does Enlarge not work?**
>
> The key to CPA is the selective mechanism, so just enlarging features does not have the attention ability. Also, direct upsampling may change the feature quality, resulting in poor performance.
>
> **Why does Attention-only just improve by 0.1%?**
>
> If we just do scale attention, the effect of the alignment and selection on a small scale is not as good as that on a large scale.
>
> We have clarified the experiment explanations in Section 4.2 and added differences between CPA and ASF in Appendix Section 6.2.

---

> ### Author Response · Authors · 2022-11-19
> **1,  The confusion about  "the representation cannot represent lines that have multiple values for single x"**
>
> The single-value property of the curve function mentioned in section 3.2 is not the limitation of our representation.
>
> The paragraph explains that a polynomial curve in the form of  y=f(x) is not suitable for representing vertical boundaries(the left and the right boundary ). To solve this problem, we utilize two kinds of functions to describe the four borders, where the top and bottom borders are with the form $y=f^{t}(x)$ and $f^{b}(x)$ as in Eq.1, while the left and right borders are with the form $x=f^{l}(y)$ and $x=f^{r}(y)$ as in Eq. 2.

---

> > ### Comment · Reviewer_FmtQ · 2022-11-25
> > **can it represent a circle?**
> >
> > Probably, my writing was confusing. I wanted to mean it cannot represent extreme cases like circles. I think it is rare to observe lines that cannot be represented by the representation, but it is still a fundamental limitation.
> >
> > Also, if you use four curves to represent a line, you will also have an issue to decide which curve is for vertical or horizontal edges for ambiguous cases (simple case is like a line rotated by 45 degree.). It could be also a weakness of the approach in practice.

---

> > > ### Author Response · Authors · 2022-11-30
> > > **PB can represent a circle**
> > >
> > > Yes, PB can represent a circle.
> > >
> > > Regarding representative ability, we can use the below PB to represent a unit circle:
> > >
> > > $y^{top}=-5.71920143e-01x^2 + 4.50009068e-04x + 1.00417641, x\in\left[-\sqrt{2}/2,\sqrt{2}/2\right]$
> > >
> > > $y^{bottom}=5.71920143e-01x^2 + -4.50009068e-04x + -1.00417641, x\in\left[-\sqrt{2}/2,\sqrt{2}/2\right]$
> > >
> > > $x^{left}=5.68226716e-01y^2 + -6.81122671e-17y + -1.00366619, y\in\left[-\sqrt{2}/2,\sqrt{2}/2\right]$
> > >
> > > $x^{right}=-5.71871125e-01y^2 + -6.59815477e-17y + 1.00416968, y\in\left[-\sqrt{2}/2,\sqrt{2}/2\right]$
> > >
> > > The Below codes can visualize that the above PB can represent a unit circle perfectly.
> > >
> > > ```
> > > import math
> > > import numpy as np
> > > import matplotlib.pyplot as plt
> > >
> > > theta = np.linspace(-np.pi, np.pi, 200)
> > > x = np.cos(theta)  # the unit circle
> > > y = np.sin(theta)  # the unit circle
> > >
> > > top_x = np.linspace(-math.sqrt(2) / 2, math.sqrt(2) / 2, 24)
> > > top_y_fit = -5.71920143e-01 * top_x ** 2 + 4.50009068e-04 * top_x + 1.00417641e+00  # top
> > > bot_x = np.linspace(-math.sqrt(2) / 2, math.sqrt(2) / 2, 24)
> > > bot_y_fit = 5.71920143e-01 * bot_x ** 2 + -4.50009068e-04 * bot_x + -1.00417641e+00  # bot
> > > left_y = np.linspace(-math.sqrt(2) / 2, math.sqrt(2) / 2, 24)
> > > left_x_fit = 5.68226716e-01 * left_y ** 2 + -6.81122671e-17 * left_y + -1.00366619e+00  # left
> > > right_y = np.linspace(-math.sqrt(2) / 2, math.sqrt(2) / 2, 24)
> > > right_x_fit = -5.71871125e-01 * right_y ** 2 + -6.59815477e-17 * right_y + 1.00416968e+00  # right
> > >
> > > fig, ax = plt.subplots(figsize=(4, 4))
> > > plt.plot(x, y, color='black', linestyle='--')
> > > plt.plot(top_x, top_y_fit, color='red', linestyle='--', linewidth=3)
> > > plt.plot(bot_x, bot_y_fit, color='green', linestyle='--', linewidth=3)
> > > plt.plot(left_x_fit, left_y, color='blue', linestyle='--', linewidth=3)
> > > plt.plot(right_x_fit, right_y, color='yellow', linestyle='--', linewidth=3)
> > >
> > > plt.show()
> > > ```
> > >
> > > As long as texts are annotated in circle shapes in the dataset, PBFormer can learn PB to fit them accurately because the PB has representative capabilities.
> > >
> > >
> > > **How about the case of a line rotated by 45 degrees?**
> > >
> > > In this case, there are no ambiguities.
> > >
> > > Firstly, we state a fact that a text is often annotated from by $N=2K$ points, where the former $K$ points are on the top of the text ($\overline{Apple}$) and the latter $K$ points are on the bottom of the text ($\underline{Apple}$). Therefore, the original annotation points already contain the **top** or **bottom** information.
> > >
> > > Then, in the case of a line rotated by 45 degrees, we can decide its four sides step-by-step, as we have introduced in Appendix Section 6.4. Here, we give a detailed procedure for this case:
> > >
> > > 1. Divide $2K$ points into four Sets.
> > >
> > > $\mathbf{S}^1=[\mathbf{p}_1,\dots,\mathbf{p}_K]$,
> > >
> > > $\mathbf{S}^2=[\mathbf{p}_{K+1},\dots,\mathbf{p}_N]$,
> > >
> > > $\mathbf{S}^3=[\mathbf{p}_K,\mathbf{p}_K+1]$,
> > >
> > > $\mathbf{S}^4=[\mathbf{p}_N,\mathbf{p}_1]$.
> > >
> > > In this case, $\mathbf{S}^1$ are points on the text's top side ($\overline{Apple}$), $\mathbf{S}^2$ are points on the text's bottom side ($\underline{Apple}$), $\mathbf{S}^3$ are points on the text's right side ($Apple|$), $\mathbf{S}^4$ are points on the text's left side ($|Apple$).
> > >
> > > 2. Assuming $\mathbf{S}^1$ and $\mathbf{S}^2$ would be fitted by a mapping $f: x \rightarrow y$, judge whether assumed functions are both single-valued. If yes, go to step 3. If not, go to step 4. In this case, assumed functions are both single-valued, so we go to step 3.
> > >
> > > 3. $\mathbf{S}^1$ and $\mathbf{S}^2$ both use the form $y=f\left(x\right)$; $\mathbf{S}^3$ and $\mathbf{S}^4$ both use the form $x=f\left(y\right)$. If $\mathbf{S}^1$'s middle point is higher than  $\mathbf{S}^2$'s, $\mathbf{S}^1$ becomes the ground truth for the top curve, and $\mathbf{S}^2$ becomes the ground truth for the bottom curve, and vice versa. If $\mathbf{S}^3$'s middle point is more left than  $\mathbf{S}^4$'s, $\mathbf{S}^3$ becomes the ground truth for the left curve, and $\mathbf{S}^4$ becomes the ground truth for the right curve, and vice versa. In this step, we can decide that: (1) the top side we need is $\mathbf{S}^1$ ($\overline{Apple}$); (2) the bottom side we need is $\mathbf{S}^2$ ($\underline{Apple}$); (3) the left side we need is $\mathbf{S}^3$ ($|Apple$); (4) the right side we need is $\mathbf{S}^4$ ($Apple|$).
> > >
> > > The above procedure for the case of a line rotated by 45 degrees has not any ambiguities.
> > >
> > > If we misunderstand the reviewer's point or the reviewer still has concerns about specific cases, we look forward to furthering engagement with the reviewer.

---

### Official Review · Reviewer_Lb4V · 2022-10-23

**Confidence:** 4
**Correctness:** 3
**Technical Novelty And Significance:** 2
**Empirical Novelty And Significance:** 2
**Recommendation:** 5

**Clarity, Quality, Novelty And Reproducibility:**

In general, the paper is clear and well written. As commented before, the whole architecture is similar to the architecture proposed in the TESTR method, with some slight variations. The main contribution is the novel text representation, although experiment do not clearly show that this representation has clear advantages over other common representations used in the past. Except for a few details (also commented before), the method could be reproduced.

**Strength And Weaknesses:**

Strengths
- The paper proposes a new compact parametric representation for text shape based on polynomials. It also defines the metric to match the points generated by the polynomial with the points in the ground-truth.
- It defines a lightweigth transformer architecture with only 2 layers with a specific coarse-to-fine strategy to generate reference points at each transformer layer.
-  A cross-pixel-attention module with no learnable parameters is introduced to perform pixel-wise attention at different feature scalses.
- The method obtains state-of-the-art results in text detection on standard datasets, with a very good efficiency in terms of computation time.

Weaknesses
- It is not clear the contribution of the new polynomial band representation. In comparison with other similar methods (for instance TESTR), it is not clear whether the improvement in the results come from the new representation or from the slight variations in the feature encoding and transformer architecture. I miss some experiment using the same architecture to different shape representations (for instance, polynomials, poitns and bezier). It is important to clarify this point as this is the main contribution of the paper. Apart from it, the architecture proposed is very similar to the one proposed in TESTR.
- In the experiments there are some results that should be better analyzed and discussed. Results for TESTR are provided without using the recognition branch in training which leads to lower results to those reported in the original TESTR paper. However, reported FPS for TESTR method in tables 1 and 2 are the values reported in the original paper using the dual branch for detection and recognition. I guess that, if only detection branch is used, FPS should be higher. For a fair comparison, FPS using only the detection branch should be reported in this case.
Furthermore, even if the method proposed here is a detection-only method, I think that a fair comparison should include results of TESTR obtained training the whole pipeline (detection and recognition). Even if the goal is only detection, It could be valid to train TESTR using the two branches and use only the detection branch to obtain the results.
- In the main paper, it is not clear in section 3.3 how the polynomial representation is matched to a ground-truth polygonal representation consisting of several points not necessarily equally distributed. In the appendix, there are some more detailes, but I think this should be made more clear in the main paper.

**Summary Of The Paper:**

The paper proposes a new text shape representation for scene text based on the coefficients of a quadratic polynomial fitting the four contours (upper, lower, left, right) of the text instance. Then, a transformer-based detector is trained to predict the coefficients of the polynomials for all text instances in an image. The transformer is designed to be lightwieght in order to improve efficiency. Experiments using standard datasets containing irregular texts show state-of-the-art results in detection accuracy and a very good efficiency in terrms of detection time.

**Summary Of The Review:**

The paper proposes a novel representation for text representation and a transformer-based architecture to predict text shapes. Although results show SoA performance, it is not clear wether the good results come from the new representation. In addition, there are some details in the comparison with the SoA that should be better presented and discussed.

---

> ### Author Response · Authors · 2022-11-19
> **1 Fair comparisons of different text representations and architectural differences to TESTR**
>
> **Adding some experiment using the same architecture to different shape representations**
>
> To validate the reviewer's concern about the contribution of polynomial curves, we conduct the experiments with the same architecture (the same as ours) but different shape representations. This Table presents ablations for various representation choices. With the same network, the performance of the polynomial band (PB) always outperforms polygon points or Bezier curves. The visualization results in appendix Appendix Section 6.6 also show that the PB can produce more accurate shapes of "distinguishing adjacent or overlapping texts".
>
> | Method | CPA | Refine-Ref. | Rep. |F. | Prec. | Rec. |
> | :----- | :-----: | :---------: | :----- |:-----: | :-----: | :-----: |
> | PBFormer | - | - | PB | **85.8** | **90.2** | **81.9** |
> | PBFormer | - | - | Pts | 82.8 | 88.4 | 77.8 |
> | PBFormer | - | - | Bez | 83.9 | 89.8 | 78.7 |
> | PBFormer | - | &check; | PB | **86.0** | **90.5** | **81.9** |
> | PBFormer | - | &check; | Pts | 83.2 | 87.6 | 79.1 |
> | PBFormer | - | &check; | Bez | 84.2 | 89.7 | 79.4 |
> | PBFormer | &check; | &check; | PB | **87.1** | **92.1** | **82.6** |
> | PBFormer | &check; | &check; | Pts | 84.6 | 90.5 | 79.5 |
> | PBFormer | &check; | &check; | Bez | 85.3 | 90.9 | 80.4 |
>
> All the reported results are also displayed in Section 4.2 and visualized in Appendix Section 6.6.
>
> To further validate the effect of PB, we also conducted experiments when comparing PBFormer-- and TESTR. Here PBFormer-- means PBFormer without proposed CPA & Refine References. This Table shows that using TESTR's network which uses two-stage Deformable-DETR and more transformer layers, PB also outperforms other output heads.
>
> | Method      | Basic Model  | #TR       | Rep. | F.     | Prec. | Rec. |
> | :-----           | :-----              | :---------: | :----- |:-----: | :-----: | :-----: |
> | PBFormer-- |Deformable-DETR              | 2      | Pts | 82.8 | 88.4 | 77.8 |
> | PBFormer-- |Deformable-DETR              | 2      | Bez | 83.9 | 89.8 | 78.7 |
> | PBFormer-- | Deformable-DETR             | 2      | PB   | **85.8** | **90.2** | **81.9** |
> | TESTR        | two-stage Deformable-DETR | 6      | Pts   | 85.3 | 89.7 | 81.2 |
> | TESTR        | two-stage Deformable-DETR | 6      | Bez  | 86.3 | 90.3 | 82.6 |
> | PBFormer--   | two-stage Deformable-DETR | 6      | PB   | **87.4** | **91.9** | **83.3** |
>
>
> **Differences from TESTR**
>
> The architecture proposed in our paper is different from the TESTR.
>
> (1) PBFormer extends single-stage Deformable-DETR without relying on intermediate bounding box results. In contrast, TESTR adopts the two-stage Deformable-DETR. TESTR predicts the bounding box in each feature point of the transformer encoder's output, then selects topK boxes based on confidence to embed them into positional embeddings and reference points.
>
> (2) PBFormer only utilizes two layers of transformer encoders and decoders. In comparison, TESTR uses six layers of transformer encoders and decoders.
>
> (3) PBFormer contains a CPA module between the backbone and transformer. The selective mechanism of CPA is compatible with DETR-like detectors that do not have non-maximum suppression. Two pieces of evidence: if we replace CPA with ASF or FPN, the performance drop; We only utilize two-layer transformer encoders and decoders, much less than the TESTR, indicating the effectiveness of the fused feature map.
>
> In Appendix Section 6.3, we summarize architectural differences compared with TESTR.

---

> ### Author Response · Authors · 2022-11-19
> **2 Update TESTR's performance and FPS**
>
> (1) We add TESTR's reported results in Table 1 and Table 2, which are trained with both detection and recognition branches and are tested detection branches,
>
> (2) We further report the speed of detection-only branch TESTR in Tables 1 and 2, costing 7.3FPS and 6.9 FPS on CTW1500 and Total-Text.
>
> TESTR achieves 87.1 and  88.0 on CTW1500 and Total-Text with both detection and recognition annotation for training, which is still less than PBFormer.
>
> Also, the FPS of TESTR's detection-only branch is also much slower than PBFormer's FPS (7.3 vs. 24.7 on CTW1500 and 6.9 vs. 24.6 on Total-Text).

---

> ### Author Response · Authors · 2022-11-19
> **3 "How PB is matched to GT polygonal points distributed not necessarily equally?" and the relation between section 3.3 and some detailes in appendix**
>
> **"How the polynomial representation is matched to a ground-truth polygonal representation consisting of several points not necessarily equally distributed?"**
>
> (1) How to calculate the loss between one ground truth polygonal representation and one PB?
>
> Firstly,
> we split the ground truth polygonal representation into four point sets, each of which contains ground truth points for one of the top, bottom, left, and right sides. In this step, no need for equal distribution. A detailed procedure is shown in Appendix Section 6.4.
>
> Secondly,
> given a top curve of PB and the GT point set for the top side, we **evenly sample**  the curve and the GT to calculate fitting losses. So does the bottom, left, and right curve. See Eq. (7).
>
> **Important**. Here, we need to resample GT to be equally distributed by Eq. (6) so that we can ensure a **one-by-one** matched correspondence of the top side's GT points and the top curve's PRED points (also equally sampled on the curve) to calculate a reasonable fitting loss.
>
> (2) How to match many PBs and many GTs for an image?
>
> The bipartite matching problem is defined in Eq. (8) and solved by the Hungarian algorithm. In Section 3.3, at the **The bipartite matching for the whole image** paragraph, we have clarified this procedure (Eq. 8-10).
>
> (3) How to calculate loss between many PBs and many GTs?
>
> With the matching result of (2), we know the one-to-one correspondence of a PB and a GT so that we can calculate each pair's fitting loss by calculating top, bottom, left, and right fitting losses by (1). This part has been clarified in the paragraph of **Overall Loss** (Eq 11-12).
>
> **"In the appendix, there are some more details, but I think this should be made more clear in the main paper."**
>
> In the appendix, we explain how we split the ground truth polygonal representation into four point sets, each of which contains ground truth points for one of the top, bottom, left, and right sides. We have added a description in Section 3.3 to refer to this step for a clear understanding.

---

### Official Review · Reviewer_cjr7 · 2022-10-24

**Confidence:** 5
**Correctness:** 3
**Technical Novelty And Significance:** 2
**Empirical Novelty And Significance:** 2
**Recommendation:** 5

**Clarity, Quality, Novelty And Reproducibility:**

The paper is well-written and well-organized. I think it is easy for an ordinary researcher to reproduce the proposed scene text detector.
However, as mentioned above, I have some concerns about the limited novelty and insufficient technical contribution of this paper.


**Strength And Weaknesses:**

Strengths:
The proposed PB text representation utilizes four polynomial curves with a fixed number of parameters to fit a text instance, while polygon-points-based methods require using different numbers of points to describe different text regions. This facilitates the learning of parameters for the predicted outline of a text region, and thus reducing the computational cost of the adopted transformer model. Extensive experiments have been conducted and the results are convincing.

Weaknesses:
1) The ABCnet [Liu et al. 2020] also adopted a fixed number of parameters to fit the outline of a text region. Therefore, what is the difference between the text representation based on Bezier curves [Liu et al. 2020] and the proposed PB base on Polynomial curves? No discussion regarding this issue has been provided. To this end, the novelty is relatively limited. The authors need to explain more at this point instead of simply claiming the PB’s advantages against polygon-points-based methods.
2) Vey few technical contributions have been made in the whole network architecture. The three modules contained in the network all utilize well-known techniques. This leads to my concerns on the insufficient academic contribution of this paper.


**Summary Of The Paper:**

This paper proposes a scene text detector, PBFormer, using the transformer with a “new” text instance representation called Polynomial Band (PB). The proposed PB is able to represent a text with a complex shape since it utilizes four polynomial curves to fit the text instance’s top, bottom, left, and right sides. The proposed PBFormer is basically a transformer model with the proposed PB text representation. Extensive experiments have been conducted to verify the effectiveness of the proposed PBFormer and its superiority to other existing approaches.

**Summary Of The Review:**

As analyzed in my above comments, there exist both clear strengths and weaknesses in this paper. To sum up, I think the quality of this paper is slightly below the bar of ICLR. I will be happy to read the author's responses and discuss with other reviewers to make my final decision.

---

> ### Author Response · Authors · 2022-11-19
> **1 The difference from the Bezier curve and experiments to validate PB's effectiveness**
>
> **The difference from the Bezier curve:**
>
> Let us consider the parameters optimized in our loss functions. For the polynomial curve, the optimized parameters are for the function defined in the image space. For the Bezier curve, the optimized parameters, \emph{i.e.}, the control points, are for the function defined with a specific variable "t".
>
> We argue that the form of the curves will finally determine how we optimize their parameters.
>
> (1) For the polynomial curve,  we can measure the difference between the annotated ground truth points and the predicted polynomial function by comparing evenly sampled points indicated polynomial curve and ground truth text-line.  Such a loss function can better reflect how humans percept the shape difference because the metric is performed on the image space.
>
> (2) For the  Bezier curve, we need first to fit the ground truth to obtain the control points by the least square method, then supervise the predicted Bezier curve by comparing the difference between the control points. Notably, the mismatch in  Bezier's control point cannot reflect the difference in the curve shape. A slight difference in the control points may bring a large curve shape change.
>
>
> We have added the above discussion in Appendix Section 6.1 to show the advantages of the polynomial curve compared with the Bezier curve. Besides, in Appendix Section 6.1, we also show why the Bezier curve cannot be learned like a polynomial curve in the image space. Learning curve by directly comparing with text-line in image space is more reasonable to reflect how humans percept shape differences.
>
>
> **The effectiveness of PB:**
>
> To validate the effect of PB, we conduct the experiments with the same architecture (the same as ours) but different shape representations.
> This Table presents ablations for various representation choices. With the same network, the performance of the polynomial band (PB) always outperforms polygon points or Bezier curves. The visualization results in appendix Appendix Section 6.6 also show that the PB can produce more accurate shapes of "distinguishing adjacent or overlapping texts".
>
>
> | Method | CPA     | Refine-Ref. | Rep.   |F.      | Prec.   | Rec.    |
> | :----- | :-----: | :---------: | :----- |:-----: | :-----: | :-----: |
> | PBFormer | - | - | PB | **85.8** | **90.2** | **81.9** |
> | PBFormer | - | - | Pts | 82.8 | 88.4 | 77.8 |
> | PBFormer | - | - | Bez | 83.9 | 89.8 | 78.7 |
> | PBFormer | - | &check; | PB | **86.0** | **90.5** | **81.9** |
> | PBFormer | - | &check; | Pts | 83.2 | 87.6 | 79.1 |
> | PBFormer | - | &check; | Bez | 84.2 | 89.7 | 79.4 |
> | PBFormer | &check; | &check; | PB | **87.1** | **92.1** | **82.6** |
> | PBFormer | &check; | &check; | Pts | 84.6 | 90.5 | 79.5 |
> | PBFormer | &check; | &check; | Bez | 85.3 | 90.9 | 80.4 |
>
> All the reported results are also displayed in Section 4.2 and visualized in Appendix Section 6.6.
>
>
> To further validate the effect of PB, we also conducted experiments when comparing PBFormer-- and TESTR. Here PBFormer-- means PBFormer without proposed CPA & Refine References. This Table shows that using TESTR's network which uses two-stage Deformable-DETR and more transformer layers, PB also outperforms other output heads.
>
> | Method        | Basic Model               | #TR | Rep. | F. | Prec. | Rec. |
> | :----- | :----- | :---------: | :----- |:-----: | :-----: | :-----: |
> | PBFormer-- |Deformable-DETR              | 2      | Pts | 82.8 | 88.4 | 77.8 |
> | PBFormer-- |Deformable-DETR              | 2      | Bez | 83.9 | 89.8 | 78.7 |
> | PBFormer-- | Deformable-DETR             | 2      | PB   | **85.8** | **90.2** | **81.9** |
> | TESTR        | two-stage Deformable-DETR | 6      | Pts   | 85.3 | 89.7 | 81.2 |
> | TESTR        | two-stage Deformable-DETR | 6      | Bez  | 86.3 | 90.3 | 82.6 |
> | PBFormer-- | two-stage Deformable-DETR   | 6      | PB   | **87.4** | **91.9** | **83.3** |

---

> > ### Comment · Reviewer_cjr7 · 2022-11-29
> > **Reply to the authors**
> >
> > I have carefully read the authors' responses. However, I still have concerns about the novelty of this paper. Compared to other papers I reviewed for ICLR 2023, the novelty of this paper is relatively low. Indeed, there exist some advantages in this paper, I think it is still slightly below the bar of the average ICLR standard.

---

> ### Author Response · Authors · 2022-11-19
> **2 The CPA module's contribution in the network architecture.**
>
> We didn't claim all three modules are the main contributions of our paper. The CPA, the PB, and the loss function are the contributions.
>
> Among the three modules.  CPA is original though it looks like ASF in DB++.
>
> (1) CPA performs scale attention in a new way: It aligns the feature maps of the different scales by enlarging the small-scale feature maps. The attention map is not obtained by introducing a new network module but by comparing the values of the existing feature maps among different scales with softmax. All the feature maps are weighted with the attention map and are resized to the original size.
>
> (2)  The scale attention is compatible with DETR-like detectors that do not have non-maximum suppression. Two pieces of evidence: if we replace CPA with ASF, the performance drop; We only utilize two-layer transformer encoders and decoders, much less than the TESTR, indicating the effectiveness of the fused feature map.
>
> A detailed comparison of ASF and CPA, including motivations, implementations, and explanations, can be found in Appendix Section 6.2. Moreover, we have clarified CPA's motivations in Section 1 and Section 3.1 and the corresponding experiments explanations in Section 4.2.

---

### Official Review · Reviewer_Tjeh · 2022-10-24

**Confidence:** 5
**Correctness:** 2
**Technical Novelty And Significance:** 2
**Empirical Novelty And Significance:** 2
**Recommendation:** 3

**Clarity, Quality, Novelty And Reproducibility:**

Clarity: The paper is well organized.
Quality: The paper needs to give more theoretical analysis.
Novelty: The proposed polynomial band is somewhat novel. Optimized fitting loss brings incremental advances.
Reproducibility: The experimental setup is well described.


**Strength And Weaknesses:**

Strength
1. The paper proposes Polynomial Band, which uses four polynomial curves to represent the text instance.
2. The optimized fitting loss is showed to be effective.
3. The method can achieve promising performance without per-training.

Weaknesses
1. The cross-scale attention is similar to Adaptive Scale Fusion in DB++. Compared with general attention operation, why can the parameter-free one highlight text regions? The author should show the differences and give more detailed explanations.
2. The motivation for CPA is not clear. It works similar to an encoder block in transformers, which may explain the use of fewer encoder blocks.
3. Compared with Bezier curve, polynomial curve is just another form of the fitting function. It seems to be more complicated in terms of the parameters and loss calculation cost. It is recommended to show the advantages more clearly.
4. The visualizations only display the increase of precision or recall, which I prefer to contribute to the strong feature extraction network rather than polynomial curves. The author better shows the superiority in "distinguishing adjacent or overlapping texts".
5. The training epoches are much larger than previous methods. The good performance may come from the stacking of computing resources.
6. To show the robustness, it is recommended to test on more challenging datasets, such as ICDAR MLT 2019, ICDAR 2019 Art and DASTA1500.


**Summary Of The Paper:**

The paper proposes a novel text representation named Polynomial Band. Cross-scale attention is used to enhance the feature of text regions. The fitting loss function is optimized to adapt to scene text detection task. The results on two benchmarks show the effectiveness.

**Summary Of The Review:**

The contributions are incremental. The analysis and explanations are not enough. It is not enough to meet the quality of ICLR.

---

> ### Author Response · Authors · 2022-11-19
> **1 The cross-scale pixel attention's differences and detailed explanations**
>
> Although CPA and ASF utilize multi-scale feature maps, they are very different in motivation and implementation.
>
>
> **Motivation Difference**:
>
> (1) ASF works with DB++, a segmentation-based method to predict the pixel-wise text region. The motivation of ASF is to help DB++ obtain a more accurate segment mask, which includes both global structures and local details.
>
> (2) CPA works with PBFormer, a detection-based method to predict the parameters of object shape. It is known that the detection results are sensitive to the scale, and the motivation of CPA is to highlight the best scale and suppress the others. Our experiments demonstrate that the selection mechanism of CPA is particularly crucial for DETR-like detectors (including our method) in text detection, which don't have non-maximum suppression(NMS) for post-processing. If we replace CPA with ASF, as in Table 3, the performance will drop 1.8% in terms of the F-score.
>
>
> **Implementation Difference**:
>
> (1) ASF performs spatial attention. It first generates the mask weights for different feature maps by a stack of Conv-ReLU and Conv-Sigmoid modules, then concatenates the mask-weighted feature maps together.
>
> (2) CPA performs scale attention. It aligns the feature maps of the different scales by enlarging the small-scale feature maps. The attention map is not obtained by introducing a new network module but by comparing the values of the existing feature maps among different scales with softmax. All the feature maps are weighted with the attention map and then resized to the original size.
>
>
> **Explanation Summary**:
>
> (1) To preserve both the global structure and local details of the segmentation mask, ASF performs spatial attention for each feature map and concatenates the information of all the feature maps.
>
> (2) To produce the detection results with an accurate scale, CPA performs scale attention among different feature maps and selectively highlights the feature maps of the best scale, and suppresses the feature maps of the other scale.
>
> (3) We assume the selective mechanism of CPA is compatible with DETR-like detectors that do not have non-maximum suppression. Two pieces of evidence: if we replace CPA with ASF, the performance drop; We only utilize two-layer transformer encoders and decoders, much less than the TESTR, indicating the effectiveness of the fused feature map.
>
>
> We have added the above discussion in Appendix Section 6.2 to show the differences between CPA and ASF. Moreover, we have clarified CPA's motivations in Section 1 and Section 3.1 and the corresponding experiments explanations in Section 4.2.

---

> ### Author Response · Authors · 2022-11-19
> **2 CPA's motivation and difference to the encoder block in transformers**
>
> (1) The motivation of CPA is explained in the 1st question. CPA performs the feature map weighting (selection) among the corresponding feature values of different scales. The selective mechanism of CPA is potentially compatible with DETR-like detectors that do not have non-maximum suppression
>
> (2) CPA and the encoder block work differently with two clear pieces of evidence: (1) CPA doesn't have parameters, while the encoder block is densely parameterized. (2) The utilization of CPA reduces both encoder blocks and decoder blocks. It doesn't just reduce s the layers of the encoder (e.g., CPA only requires two enc and two dec, while TESTR has six enc and six dec).

---

> ### Author Response · Authors · 2022-11-19
> **3 The polynomial curve's advantages compared to Bezier curve**
>
> Yes, the polynomial curve is another form of the fitting function, but a better form.
>
> Let us consider the parameters optimized in our loss functions. For the polynomial curve, the optimized parameters are for the function defined in the image space. For the Bezier curve, the optimized parameters, \emph{i.e.}, the control points, are for the function defined with a specific variable "t".
>
> **Important**: The form of the curves will finally determine how we optimize their parameters.
>
> (1) For the polynomial curve,  we can measure the difference between the annotated points and the predicted polynomial function by comparing evenly sampled indicated polynomial curve and ground truth text-line.
>
> Such a metric can better reflect how humans percept the shape difference.
>
> (2) For the  Bezier curve, we need first to fit the ground truth to obtain the control points, then supervise the predicted Bezier curve by comparing the difference between the control points. Notably, the mismatch in  Bezier's control point cannot reflect the difference in the curve shape.  A slight difference in the control points may bring a large shape change.
>
> We have added the above discussion in Appendix Section 6.1 to show the advantages of the polynomial curve compared with the Bezier curve. Besides, in Appendix Section 6.1, we also show why the Bezier curve **cannot** be learned like a polynomial curve in the image space. Learning curve by directly comparing it with text-line in image space is more reasonable to reflect how humans percept shape differences.

---

> ### Author Response · Authors · 2022-11-19
> **4 Validate the effectiveness of polynomial curves and show visualizations for adjacent or overlapping texts**
>
> The visualization results in appendix Appendix Section 6.6 also show that the PB can produce more accurate shapes of "distinguishing adjacent or overlapping texts".
>
> To validate the reviewer's concern about the contribution of polynomial curves, we conduct the experiments with the same architecture (the same as ours) but different shape representations. This Table presents ablations for various representation choices. With the same network, the performance of the polynomial band (PB) always outperforms polygon points or Bezier curves.
>
>
>
> | Method  | CPA | Refine-Ref.| Rep.     |  F.      |   Prec.      | Rec.   |
> | :----: | :---: | :-------: | :----:  | :----: |  :----: | :----: |
> | PBFormer  | - | - | PB | **85.8** | **90.2** | **81.9** |
> | PBFormer | - | - | Pts | 82.8 | 88.4 | 77.8 |
> | PBFormer | - | - | Bez | 83.9 | 89.8 | 78.7 |
> | PBFormer | - | &check; | PB | **86.0** | **90.5** | **81.9** |
> | PBFormer | - | &check; | Pts | 83.2 | 87.6 | 79.1 |
> | PBFormer | - | &check; | Bez | 84.2 | 89.7 | 79.4 |
> | PBFormer | &check; | &check; | PB | **87.1** | **92.1** | **82.6** |
> | PBFormer | &check; | &check; | Pts | 84.6 | 90.5 | 79.5 |
> | PBFormer | &check; | &check; | Bez | 85.3 | 90.9 | 80.4 |
>
>
> All the reported results are also displayed in Section 4.2 of the revised manuscript to analyze the proposed components.
>
> To further validate the effect of PB, we also conducted experiments when comparing PBFormer-- and TESTR. Here, PBFormer-- means PBFormer without proposed CPA & Refined References. This Table shows that using TESTR's network, which uses two-stage Deformable-DETR and more transformer layers, PB also outperforms other output heads.
>
> | Method        | Basic Model     | #TR | Rep. | F. | Prec. | Rec. |
> | :---- | :--- | :-------: | :----:  | :----: |  :----: | :----: |
> | PBFormer-- |Deformable-DETR      | 2      | Pts | 82.8 | 88.4 | 77.8 |
> | PBFormer-- |Deformable-DETR      | 2      | Bez | 83.9 | 89.8 | 78.7 |
> | PBFormer-- | Deformable-DETR     | 2      | PB   | **85.8** | **90.2** | **81.9** |
> | TESTR      | two-stage Deformable-DETR | 6      | Pts   | 85.3 | 89.7 | 81.2 |
> | TESTR        | two-stage Deformable-DETR | 6      | Bez  | 86.3 | 90.3 | 82.6 |
> | PBFormer-- | two-stage Deformable-DETR | 6      | PB   | **87.4** | **91.9** | **83.3** |

---

> ### Author Response · Authors · 2022-11-19
> **5 "Training epochs are large and the good performance may come from the stacking of computing resources"**
>
> (1) We disagree with the reviewer's viewpoint that good performance can come from large training epochs. It is known that the performance of deep neural networks is usually saturated under long-time training, and the current papers often report the results until convergence.
>
>
> (2) As reported in the paper, " The training process takes about 2 days on 4 Tesla V100 GPUs with the image batch size of 14."  We don't think such an amount of computation can be called ``the stacking of computing resources" in today's community.
>
>
> (3) Another transformer-based text detector, TESTR, fine-tuned CTW1500 with 200k iterations, 8 GPUs, and 8 batch sizes, resulting in 8 * 8 * 200k / 1k = 12800 epochs. In comparison, we use 9000 epochs for training, which is 25% more efficient.

---

> ### Author Response · Authors · 2022-11-19
> **6 Experiments on DAST1500, MLT2019 and Art2019**
>
> We guess the reviewer recommends we test on DAST1500. We follow "Mask is All You Need: Rethinking Mask R-CNN for Dense and Arbitrary-Shaped Scene Text Detection [MM'21]" to conduct experiments. PBFormer achieves 85.9%, which is comparable with the previous state-of-the-art "Mask-RCNN + ALA + MMD (MAYOR)".
>
> | Method      | F. | Prec. | Rec.|
> | :---- | :---: | :-------: | :----:  |
> | TextBoxes      | 50.9 | 67.3 | 40.9 |
> | RRD            | 53.0 | 67.2 | 43.8 |
> | EAST           | 62.0 | 70.0 | 55.7 |
> | SegLink        | 65.3 | 66.0 | 64.7 |
> | CTD+TLOC       | 66.6 | 73.8 | 60.8 |
> | PixelLink      | 74.7 | 74.5 | 75.0 |
> | ICG            | 79.4 | 79.6 | 79.2 |
> | ReLaText       | 85.8 | 89.0 | 82.9 |
> | MAYOR          | 86.6 | 87.8 | **85.5** |
> | PBFormer       | 85.9 | 90.0 | 82.1 |
> | PBFormer (two-stage)  | **86.7** | **90.4** | 83.2 |
>
> Discussion: (1) DAST1500 is full of dense texts. MAYOR is two-stage and has spatial designs for those cases, such as region proposal network (RPN) and MLP Mask Decoder (MLM).
>
> (2) PBformer is single-stage. Although it performs less well than MAYOR, it is more efficient because it does not have post-processing with NMS. If we take training strategy same with two-stage Deformable-DETR, we can achieve better performance (86.7 vs. 86.6). Besides, one-stage PBFormer still outperforms MAYOR on CTW1500 (87.0 vs. 85.3) and Total-Text (87.1 vs. 86.3).
>
> Methods on leaderboards of ICDAR MLT 2019 and ICDAR ArT 2019 adopt ResNeSt-101, ResNeXt-101, or ResNext-152 as their backbones which are far heavy for a fair comparison.
>
> Using a similar size network based on ResNet50, we conduct experiments across DB++, FCENet, TESTR, and PBFormer by splitting the training set of MLT2019 and Art2019 into training and validation with 1:1. We generate ground truth of Bezier control points following ABCNet.
>
> | Method    | Rep. | F.MLT2019 | F.Art2019 |
> | :---- | :---: | :-------: | :----:  |
> | DB++       | Seg | 64.7 | 71.1 |
> | FCENet    | Seg | 66.3 | 74.0 |
> | TESTR     | Pts | 68.7 | 75.2 |
> | TESTR     | Bez | 66.7 | 73.6 |
> | PBFormer  | PB | **69.3** | **76. 2** |
>
> All the reported results and **visualizations** are displayed in Appendix Section 6.5.

---

### Author Response · Authors · 2022-11-19
**General Response**

**We previously put the appendix into the supplementary materials. According to the responses from reviewers, we feel that the appendix may be ignored by several reviewers. Now we add the appendix in the manuscript after the reference.**

We thank the reviewer for the insight and feedback. The reviewers noted that:
(1) adding experiments using different text representations with the same network and other datasets' performances;
(2) differences about the proposed polynomial band and CPA module compared with the Bezier curve and multi-scale modules, respectively;
(3) descriptions of some experimental results and implementation details.

We addressed all concerts in the individual comments. Also, we list a summary of the changes we made to the paper according to reviewers' comments.

(1) Polynomial band's advantages compared with Bezier curve in Section 1.

(2) Descriptions to clarify CPA's motivations in Section 1 and Section 3.1.

(3) Descriptions refer to the part about how to split a ground truth polygon into four point sets for four sides in Section 3.3.

(4) Update TESTR's' results in Table 1, Table 2, and Section 4.1.

(5) Ablation study on the proposed polynomial band and CPA in Section 4.2.

Moreover, we have changed the appendix by adding the following clarifications, experiments, and visualizations according to the reviewers' recommendations.

(1) Detailed comparisons between the polynomial curve and the Bezier curve in Appendix Section 6.1;

(2) Differences between the proposed CPA and the previous ASF in Appendix Section 6.2;

(3) Experiments results on DAST1500, MLT2019 and Art2019 in Appendix Section 6.5;

(4) Visualized comparisons of different text representations with the same network for adjacent and overlapping texts in Appendix Section 6.6.


We hope we have addressed the concerns and questions raised by reviewers in a satisfactory and elucidatory manner, and we look forward to furthering engagement with the reviewer.

---

### Decision · Program_Chairs · 2023-01-20

**Decision:**

Reject

**Justification For Why Not Higher Score:**

There simply isn't much reviewer enthusiasm about this paper. The proposed method works, and it beats prior work according to the metrics shown, but the novel insights that are offered don't rise to the level of significance that would merit an ICLR publication.

**Justification For Why Not Lower Score:**

N/A

My "This decision can be bumped up" below means that I am open to the SAC having a different opinion about the magnitude of contribution this paper offers and whether it would be of interest to ICLR.

**Metareview: Summary, Strengths And Weaknesses:**

This paper proposes a new representation for detecting text in images based on a type of polynomial curve.

Strengths:
- The proposed representation uses a fixed number of parameters for four polynomial curves, rater than variable number of parameters for prior polygon based methods
- Extensive experiments
- Good results

Weaknesses:
- Unclear advantage of polynomial curves over prior work (ABCNet) that uses Bézier curves
- Need to evaluate on more challenging datasets (the authors did this in a revision, and the results look good for them)
- Concern that better performance could be just due to the proposed method being trained for longer than comparison conditions (authors rebut this pretty convincingly)
- Unclear whether performance benefits are because of architectural changes or new polynomial curve representation (authors have provided new experiments which show this).

Reviewers had a lot of their concerns addressed by the rebuttal. Two upgraded their ratings to borderline accept...but they were not enthusiastic about accepting the paper (more like they wouldn't stand against it if someone else championed it).

Given the uncertain reviewer stances, I felt that a meeting would be necessary.

**Summary Of Ac-Reviewer Meeting:**

Reviewers felt that, while the paper seemed technically sound and achieved good performance on the metrics it targeted, the magnitude of contribution seems small (relative to prior work that already uses some form of polynomial curve, e.g. Bézier curves, for the same task). In a synchronous meeting between the AC and one of the reviewers, the reviewer expressed doubt about whether the paper offered generalizable insights that others in the ICLR community would build upon in future work. In their opinion, the paper could be acceptable at a conference that specialized in text detection but did not offer enough to the broader machine learning community to warrant publication at ICLR.